# PAC Prediction Sets for Meta-Learning

**Sangdon Park**
School of Cybersecurity and Privacy
Georgia Institute of Technology
sangdon@gatech.edu

**Edgar Dobriban**
Dept. of Statistics & Data Science
The Wharton School
University of Pennsylvania
dobriban@wharton.upenn.edu

**Insup Lee**
Dept. of Computer & Info. Science
PRECISE Center
University of Pennsylvania
lee@cis.upenn.edu

**Osbert Bastani**
Dept. of Computer & Info. Science
PRECISE Center
University of Pennsylvania
obastani@seas.upenn.edu

## Abstract

Uncertainty quantification is a key component of machine learning models targeted at safety-critical systems such as in healthcare or autonomous vehicles. We study this problem in the context of meta learning, where the goal is to quickly adapt a predictor to new tasks. In particular, we propose a novel algorithm to construct *PAC prediction sets*, which capture uncertainty via sets of labels, that can be adapted to new tasks with only a few training examples. These prediction sets satisfy an extension of the typical PAC guarantee to the meta learning setting; in particular, the PAC guarantee holds with high probability over future tasks. We demonstrate the efficacy of our approach on four datasets across three application domains: mini-ImageNet and CIFAR10-C in the visual domain, FewRel in the language domain, and the CDC Heart Dataset in the medical domain. In particular, our prediction sets satisfy the PAC guarantee while having smaller size compared to other baselines that also satisfy this guarantee.

## 1 Introduction

Uncertainty quantification is a key component for safety-critical systems such as healthcare and robotics, since it enables agents to account for risk when making decisions. Prediction sets are a promising approach, since they provide theoretical guarantees when the training and test data are i.i.d. [1, 2, 3, 4]; there have been extensions to handle covariate shift [5, 6, 7] and online learning [8].

We consider the meta learning setting, where the goal is to adapt an existing predictor to new tasks using just a few training examples [9, 10, 11, 12, 13]. Most approaches follow two steps: meta learning (*i.e.,* learn a predictor for fast adaptation) and adaptation (*i.e.,* adapt this predictor to the new task). While there has been work on prediction sets in this setting [14], a key shortcoming is that their guarantees do not address some important needs in meta learning; in particular, they do not hold with high probability over future tasks. For instance, in autonomous driving, the future tasks might be new regions, and correctness guarantees ought to hold for *each* region; in healthcare, future tasks might be new patient cohort, and correctness guarantees ought to hold for *each* cohort.

We propose an algorithm for constructing probably approximately correct (PAC) prediction sets for the meta learning setting. As an example, consider learning an image classifier for self-driving cars as our use case (Figure 1). Labeled images for image classifier learning are collected, with different regions representing different but related tasks (*e.g.,* New York, Pennsylvania, and California). The

36th Conference on Neural Information Processing Systems (NeurIPS 2022).

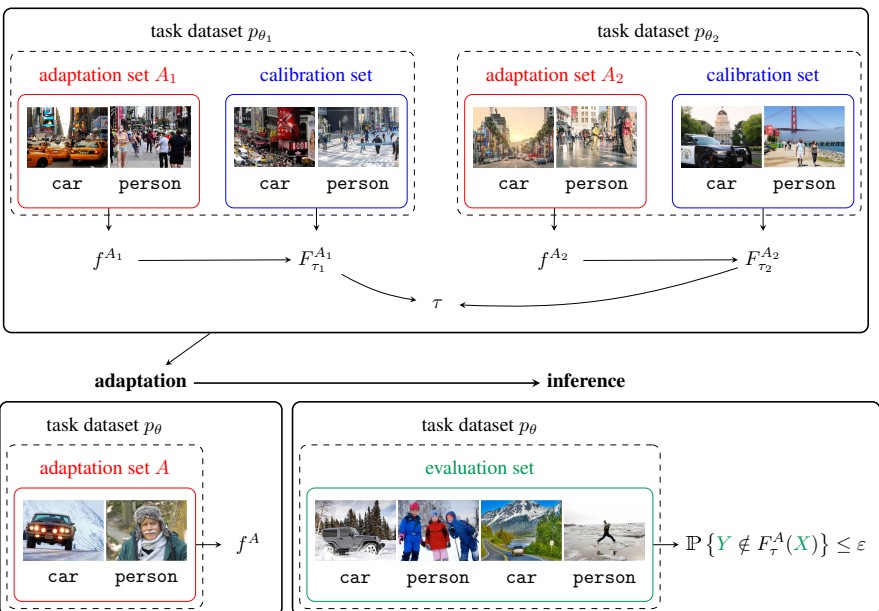

Figure 1: Learning for PAC prediction sets with fast adaptation. In meta learning and calibration, the PAC prediction set parameter $\tau$ is chosen from multiple task datasets given a meta-learned and adapted score function $f^{A_i}$ using any meta-learning algorithm and an adapted PAC prediction set $F_{\tau_i}^{A_i}$ (*e.g.,* an image classifier and a prediction set are learned over labeled images from various regions, including New York). In adaptation, the score function is adapted to the new task distribution $f^A$ (*e.g.,* the image classifier is adapted using labeled images from Alaska). In inference, by using the same parameter $\tau$ used without additional calibration, the constructed and adapted prediction set $F_{\tau}^A$ has a correctness guarantee for most of future examples (*e.g.,* the prediction set along with the adapted image classifier has a correctness guarantee over most images from Alaska). Importantly, the correctness guarantee (error below $\varepsilon$) is achieved with probability at least (1) $1 - \delta$ over the calibration sets in the meta learning phase and (2) $1 - \alpha$ over an adaptation set in the adaptation phase. This explains why we use a three-parameter PAC-style guarantee for meta learning.

goal is to deploy this learned image classifier to a new target region (*e.g.,* Alaska) in a cost-efficient way, *e.g.,* minimizing label gathering cost on labeled images from Alaska for the classifier adaptation.

Due to the safety-critical nature of the task, it is desirable to quantify uncertainty in a way that provides rigorous guarantees, where the guarantees are adapted to a new task (*e.g.,* Alaska images)— i.e., they hold with high probability specifically for future observations from this new task. In contrast, existing approaches only provide guarantees on average across future tasks [14].

Our proposed PAC prediction set starts with a meta-learned and potentially adaptable score function as our base predictor (using a standard meta learning or few shot classification approach). Then, we learn the parameter of a prediction set for meta "calibration" by using calibration task distributions (*e.g.,* labeled images from Texas and Massachusetts). Given a test task distribution (*e.g.,* labeled images from Alaska), we only have a few labeled examples for adapting the score function to this target task without further updating the prediction set parameter, which potentially saves additional labeled examples. We prove that the proposed algorithm satisfies the PAC guarantee tailored to the meta learning setup. Additionally, we demonstrate the validity of the proposed algorithm over four datasets across three application domains: mini-ImageNet [10] and CIFAR10-C [15] in the visual domain, FewRel [16] in the language domain, and CDC Heart Dataset [17] in the medical domain.

## 2  Related Work

**Meta learning for few-shot learning.** Meta-learning follows the paradigm of learning-to-learn [9], usually formulated as few-shot learning: learning or adapting a model to a new task or distribution

by leveraging a few examples from the distribution. Existing approaches propose neural network predictors or adaptation algorithms [10, 11, 12, 13] for few-shot learning. However, there is no guarantee that these approaches can quantify uncertainty properly, and thus they do not provide guarantees for probabilistic decision-making.

**Conformal prediction.** The classical goal in conformal prediction (and inductive conformal prediction) [18, 2, 19] is to find a prediction set that contains the label with a given probability, with respect to randomness in both calibration examples and a test example, where a score function (or non-conformity score) is given for the inductive conformal prediction. This property is known as marginal coverage. If the labeled examples are independent and identically distributed (or more precisely exchangeable), prediction sets constructed via inductive conformal prediction have marginal coverage [19, 2]. However, the identical distribution assumption can fail if the covariates of a test datapoint are differently distributed from the training covariates, *i.e.,* covariate shift occurs. Assuming that the likelihood ratio is known, the weighted split conformal prediction set [5] satisfies marginal coverage under covariate shift. So far, we have considered only two distributions, a calibration distribution and test distribution. In meta learning, a few examples from multiple distributions are given, providing a different setup. Assuming exchangeability of these distributions, conformal prediction sets for meta learning have been constructed, covering over a random sample from a new randomly drawn distribution, exchangeable with the training distributions [14]. In statistics, two-layer hierarchical models consider examples drawn from multiple distributions, while assuming examples and distributions are independent and identically distributed; these are identical to the model of meta-learning we study. Several approaches to construct prediction sets (*e.g.,* double conformal prediction) satisfy the marginal coverage guarantee [20].

**PAC prediction sets.** Marginal coverage is unconditional over calibration data, and does not hold conditionally over calibration data (*i.e.,* with specified probability for a fixed calibration set). *Training conditional conformal prediction* was introduced as a way to construct prediction sets with PAC guarantees [21]; Specifically, for independent and identically distributed calibration examples, they construct a prediction set that contains a true label with high probability with respect to calibration examples; thus, these prediction sets cover the labels for most future test examples. This approach applies classical techniques for constructing tolerance regions [1, 22] to the non-conformity score to obtain these guarantees. Recent work casts this technique in the language of learning theory [3], informing our extension to the meta-learning setting; thus, we adopt this formalism in our exposition. PAC prediction sets can be constructed in other learning settings—*e.g.,* under covariate shift (assuming the distributions are sufficiently smooth [6] or well estimated [7]). In the meta learning setting, the meta conformal prediction set approach also has a PAC property [14], but does not control error over the randomness in the adaptation test examples. In contrast, we propose a prediction set satisfying a *meta-PAC* property—*i.e.,* separately conditional with respect to the calibration data and the test adaptation examples; see Section 4.1 for details.

## 3 Background

### 3.1 PAC Prediction Sets

We describe a training-conditional inductive conformal prediction strategy for constructing probably approximately correct (PAC) prediction sets [21], using the reformulation as a learning theory problem in [3]. Let $\mathcal{X}$ be a set of examples or features and $\mathcal{Y}$ be a set of labels. We consider a given score function $f : \mathcal{X} \times \mathcal{Y} \to \mathbb{R}_{\geq 0}$, which is higher if $y \in \mathcal{Y}$ is the true label for $x \in \mathcal{X}$. We consider a parametrized prediction set; letting $\tau \in \mathbb{R}_{\geq 0}$, we define a prediction set $F_\tau : \mathcal{X} \to 2^{\mathcal{Y}}$ as follows:

$$F_\tau(x) \coloneqq \{y \in \mathcal{Y} \mid f(x, y) \geq \tau\}.$$

Consider a family of distributions $(p_\theta)_{\theta \in \Theta}$ over $\mathcal{Z} \coloneqq \mathcal{X} \times \mathcal{Y}$, parametrized by $\theta \in \Theta$. For an unknown $\theta$, we observe independent and identically distributed (i.i.d.) calibration datapoints $S \sim p_\theta^n$. We define a prediction set, or equivalently the corresponding parameter $\tau$, to be $\varepsilon$-correct, as follows:

**Definition 1.** A parameter $\tau \in \mathbb{R}_{\geq 0}$ is $\varepsilon$-*correct* for $p_\theta$ if

$$\mathbb{P}\left\{Y \in F_\tau(X)\right\} \geq 1 - \varepsilon, \tag{1}$$

where the probability is taken over $(X, Y) \sim p_\theta$.

Letting $T_\varepsilon(\theta)$ be the set of $\tau$ satisfying (1) for $p_\theta$, we can write $\tau \in T_\varepsilon(\theta) \subseteq \mathbb{R}_{\geq 0}$ if $\tau$ is $\varepsilon$-correct for $p_\theta$. If an estimator of $\tau$ is $\varepsilon$-correct on most random calibration sets $S \sim p_\theta^n$, we say it is PAC.

**Definition 2.** *An estimator* $\hat{\gamma}_{\varepsilon,\delta} : \mathcal{Z}^n \to \mathbb{R}_{\geq 0}$ *is* $(\varepsilon, \delta)$*-probably approximately correct (PAC) for* $p_\theta$ *if*

$$\mathbb{P}\left\{\hat{\gamma}_{\varepsilon,\delta}(S) \in T_\varepsilon(\theta)\right\} \geq 1 - \delta,$$

*where the probability is taken over* $S$.

Along with the PAC guarantee, the prediction set should be small in size. Here, by increasing the scalar parameterization, we see that the expected prediction set size is decreasing with an increasing prediction set error (*i.e.,* $\mathbb{P}_{X,Y}\{Y \notin F_\tau(X)\}$). Thus, a practical $(\varepsilon, \delta)$-PAC algorithm that implements the estimator $\hat{\gamma}_{\varepsilon,\delta}$ maximizes $\tau$ while satisfying a PAC constraint to minimize the expected prediction set size [3].

## 3.2 Meta Learning

Meta learning aims to learn a score function that can be conveniently adapted to a new distribution, also called a *task*. Consider a *task distribution* $p_\pi$ over $\Theta$, and $M$ training task distributions $p_{\theta_1}, \ldots, p_{\theta_M}$, where $\theta_1, \ldots, \theta_M$ is an i.i.d. sample from $p_\pi$. Here, we mainly consider meta learning with adaptation (*e.g.,* few-shot learning) to learn the score function, while having learning without adaptation (*e.g.,* zero-shot learning) as a special case of our main setup.

For meta learning without adaptation, labeled examples are drawn from each training task distribution to learn the score function $f^\emptyset : \mathcal{X} \times \mathcal{Y} \to \mathbb{R}_{\geq 0}$, and the same score function is used for inference (*i.e.,* at test time). For meta learning with adaptation, $t$ adaptation examples are also drawn from each training task distribution to learn the score function $f : \mathcal{Z}^t \times \mathcal{X} \times \mathcal{Y} \to \mathbb{R}_{\geq 0}$, and the same number of adaptation examples are drawn from a new test task distribution for adaptation. Given an adaptation set $A \in \mathcal{Z}^t$, the adapted score function is denoted by $f^A : \mathcal{X} \times \mathcal{Y} \to \mathbb{R}_{\geq 0}$, where $f^A(x, y) \coloneqq f(A, x, y)$. We consider $\mathcal{Y}$ large enough to include all label classes, some of which possibly seen during testing but not during training. This formulation includes learning setups (*e.g.,* few-shot learning) where unseen label classes are observed during inference, by considering different distributions over labeled examples during training and inference.

## 4 Meta PAC Prediction Sets

We propose a PAC prediction set for meta learning. In particular, we consider a meta-learned score function, which is typically trained to be a good predictor after adaptation to the test distribution. Our prediction set has a correctness guarantee for future test distributions.

There are three sources of randomness: (1) an adaptation set and a calibration set for each calibration task, (2) an adaptation set for a new task, and (3) an evaluation set for the same new task. We control the three sources of randomness and error by $\delta$, $\alpha$, and $\varepsilon$, respectively, aiming to construct a prediction set that contains labels from a new task. Given a meta-learned model, the proposed algorithm consists of two steps: constructing a prediction set—captured by a specific parameter to be specified below—for each calibration task (related to $\varepsilon$ and $\alpha$), and then constructing a prediction interval over parameters (related to $\alpha$ and $\delta$).

Suppose we have prediction sets for each task, parametrized by some threshold parameters. Suppose that, for each task, the parameters only depend on an adaptation set for the respective task. As tasks are i.i.d., the parameters also turn out to be i.i.d.. The distribution of parameters represents a possible range of the parameter for a new task. In particular, if for each task, the prediction set contains the true label a fraction $1 - \varepsilon$ of the time, the distribution of the parameters suggests how one may achieve $\varepsilon$-correct prediction sets for a new task. Based on this observation, we construct a prediction interval that contains a fraction $1 - \alpha$ of $\varepsilon$-correct prediction set parameters. However, the parameters of prediction sets need to be estimated from calibration tasks, and the estimated parameters depend on the randomness of calibration tasks. Thus, we construct a prediction interval over parameters which accounts for this randomness. The interval is chosen to be correct (*i.e.,* to contains $\varepsilon$-correct prediction set parameters for a fraction $1 - \alpha$ of new tasks) with probability at least $1 - \delta$ over the randomness of calibration tasks.

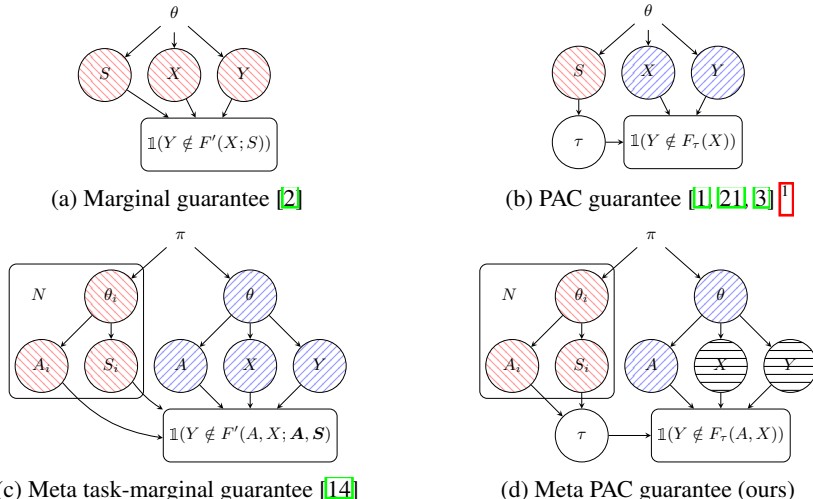

(a) Marginal guarantee [2]   (b) PAC guarantee [1, 21, 3] [1]

(c) Meta task-marginal guarantee [14]   (d) Meta PAC guarantee (ours)

Figure 2: Comparison among different guarantees on prediction sets. An edge $A \to B$ between random variables $A, B$ represents that $B$ is drawn conditionally on $A$, and a plate with the symbol $N$ represents that there are $N$ copies of the variables in it, indexed by $i = 1, \ldots, N$. Each stripe pattern with a different color indicates that the corresponding random variables are simultaneously conditioned on in the correctness guarantee; $F'$ denotes a conformal prediction set. Thus, for instance, in (d), there is a $1 - \varepsilon$ probability of correctness with respect to the test datapoint $(x, y)$, a $1 - \alpha$ probability of correctness with respect to the test task and its adaptation data $(\theta, A)$, and a $1 - \delta$ probability of correctness with respect to the calibration tasks, their calibration data, and their adaptation data $(\theta_i, S_i, A_i)$ for $i = 1, \ldots, N$. The proposed meta PAC prediction set (2d) has the most fine-grained control over the correctness guarantee conditioning over calibration tasks, the test task and adaptation set, and the test data.

## 4.1 Setup and Problem

We describe the problem on PAC prediction set construction for meta learning with adaptation; see Appendix A for a simpler problem setup without adaptation, a special case of our main problem.

**Setup.** In the setting of Section 3.2, the training samples from some training distributions $p_{\theta'_i}$, $i = 1, \ldots, M$ are used to learn a score function via an *arbitrary* meta learning method, including metric-based meta-learning [11] and optimization-based meta-learning [12]. We denote the given meta-learned score function by $f$. Additionally, we have $N$ calibration distributions $p_{\theta_1}, \ldots, p_{\theta_N}$, where $\theta_1, \ldots, \theta_N$ is an i.i.d. sample from $p_\pi$, *i.e.,* $\boldsymbol{\theta} \coloneqq (\theta_1, \ldots, \theta_N) \sim p_\pi^N$. From each calibration distribution $p_{\theta_i}$, we draw $n$ i.i.d. labeled examples for a calibration set, *i.e.,* $\boldsymbol{S} \coloneqq (S_1, \ldots, S_N) \sim p_{\theta_1}^n \cdots p_{\theta_N}^n$ and $\boldsymbol{S} \in \boldsymbol{\mathcal{S}} \coloneqq \mathcal{Z}^{n \times N}$. Additionally, we draw $t$ i.i.d. labeled examples for an adaptation set, *i.e.,* $\boldsymbol{A} \coloneqq (A_1, \ldots, A_N) \sim p_{\theta_1}^t \cdots p_{\theta_N}^t$ and $\boldsymbol{A} \in \boldsymbol{\mathcal{A}} \coloneqq \mathcal{Z}^{t \times N}$; we use them to construct the meta PAC prediction set. Finally, we consider a test distribution $p_\theta$, where $\theta \sim p_\pi$. From it, we draw i.i.d. labeled adaptation examples $A$, *i.e.,* $A \sim p_\pi^t$, and i.i.d. evaluation examples.

**Problem.** Our goal is to find a prediction set $F_\tau : \mathcal{Z}^t \times \mathcal{X} \to 2^{\mathcal{Y}}$ parametrized by $\tau \in \mathbb{R}_{\geq 0}$ which satisfies a meta-learning variant of the PAC definition. In particular, consider an estimator $\hat{\tau} : \boldsymbol{\mathcal{S}} \times \boldsymbol{\mathcal{A}} \to \mathbb{R}_{\geq 0}$ that estimates $\tau$ for a prediction set, given an *augmented calibration set* $(\boldsymbol{S}, \boldsymbol{A})$. Let $T_\varepsilon(\theta, A)$ be a set of all $\varepsilon$-correct parameters $\tau$ with respect to $f$ adapted using a test adaptation set $A$. We also consider $F_\tau^A(x, y) \coloneqq F_\tau(A, x, y)$.

We say that a prediction set $F_\tau$ parametrized by $\tau$ is $\varepsilon$-correct for $p_\theta$ for a fixed adaptation set $A$, if $\tau \in T_\varepsilon(\theta, A)$. Moreover, we want the prediction set $F_\tau$ where $\tau$ is estimated via $\hat{\tau}$ to be correct on most test sets $A$ for adaptation and for most test tasks $\theta$, *i.e.,* given $\boldsymbol{S} \in \boldsymbol{\mathcal{S}}$ and $\boldsymbol{A} \in \boldsymbol{\mathcal{A}}$,

$$\mathbb{P}\left\{ \hat{\tau}(\boldsymbol{S}, \boldsymbol{A}) \in T_\varepsilon(\theta, A) \right\} \geq 1 - \alpha,$$

where the probability is taken over $\theta$ and $A$. Intuitively, we want a prediction set, adapted via $A$, to be correct on future labeled examples from $p_\theta$. Finally, we want the above to hold for most augmented calibration sets $(\boldsymbol{S}, \boldsymbol{A})$ from most calibration tasks $\theta_1, \ldots, \theta_N$, leading to the following definition.

**Definition 3.** *An estimator* $\hat{\tau} : \boldsymbol{S} \times \boldsymbol{A} \to \mathbb{R}_{\geq 0}$ *is* $(\varepsilon, \alpha, \delta)$-*meta probably approximately correct* (mPAC) *for* $p_\pi$ *if*

$$\mathbb{P}\left\{ \mathbb{P}\left\{ \hat{\tau}(\boldsymbol{S}, \boldsymbol{A}) \in T_\varepsilon(\theta, A) \right\} \geq 1 - \alpha \right\} \geq 1 - \delta,$$

*where the outer probability is taken over* $\boldsymbol{\theta}, \boldsymbol{S}$ *and* $\boldsymbol{A}$, *and the inner probability is taken over* $\theta$ *and* $A$.

Notably, Definition 3 is tightly related to the PAC definition of meta conformal prediction [14]. We illustrate the difference in Figure 2; the figure represents the different guarantees of prediction sets via graphical models. Each stripe pattern indicates the random variables that are conditioned on in the respective guarantee. Figure 2d represents the proposed guarantee, having Figure 2b and 7b as special cases. Figure 2c shows the guarantee of meta conformal prediction proposed in [14], also having Figure 2a and 7a as special cases. Figure 2d shows that the proposed correctness guarantee is conditioned over *calibration tasks*, an *adaptation set A from a test task*, and *test data* $(x, y)$. This implies that the correctness guarantee of an adapted prediction set (thus conditioned on the adaptation set) holds on most future test datasets from the same test task. In contrast, the guarantee in Figure 2c is conditioned on the entire test task. Thus, the correctness guarantee does not hold conditionally over adaptation sets; this means that the adapted prediction set (thus conditioned on an adaptation set) does not need to be correct over most future datasets in the same task.

## 4.2 Algorithm

---

**Algorithm 1 Meta-PS**: PAC prediction set for meta-learning. Internally, any PAC prediction set algorithms are used to implement an estimator $\hat{\gamma}_{\varepsilon, \delta}$; we use PS-BINOM (Algorithm 2) in Appendix B.

---

1: **procedure** META-PS($\boldsymbol{S}, \boldsymbol{A}, f, g, \varepsilon, \alpha, \delta$)
2:      **for** $i \in \{1, \ldots, N\}$ **do**
3:          $\tau_i \leftarrow$ PS-BINOM($S_i, f^{A_i}, \varepsilon, \alpha/2$)          ($\triangleright$) Use Algorithm 2 in Appendix B
4:      $S' \leftarrow ((\tau_1, 1), \ldots, (\tau_N, 1))$
5:      $\tau \leftarrow$ PS-BINOM($S', g, \alpha/2, \delta$)
6:      **return** $\tau$

---

Next, we propose our meta PAC prediction set algorithm for an estimator $\hat{\tau}$ that satisfies Definition 3 while aiming to minimize the prediction set size, by leveraging the scalar parameterization of a prediction set. The algorithm consists of two steps: find a prediction set parameter over labeled examples $S_i$ and $A_i$ from the $i$-th calibration task distribution, and find a prediction set over the parameter of prediction sets in the previous step. At inference, the constructed prediction set is adapted to a test task distribution via an adaptation set $A$. Importantly, in finding prediction sets (*i.e.,* implementating an $(\varepsilon, \delta)$-PAC estimator $\hat{\gamma}_{\varepsilon, \delta}$), we can use any PAC prediction set algorithms (*e.g.,* Algorithm 2 in Appendix B).

**Meta calibration.** For the first step, for each $i \in \{1, \ldots, N\}$, we estimate a prediction set parameter $\hat{\gamma}_{\varepsilon, \alpha/2}(S_i)$, satisfying a $(\varepsilon, \alpha/2)$-PAC property using a calibration set $S_i$ and a score function $f$ adapted via an adaptation set $A_i$, leading to $f^{A_i}$, by using any meta learning and adaptation algorithm. We consider the $N$ learned prediction set parameters—with a dummy label 1 to fit the convention of the PAC prediction set algorithm—to form a new labeled sample, *i.e.,* $S' := ((\hat{\gamma}_{\varepsilon, \alpha/2}(S_1), 1), \ldots, (\hat{\gamma}_{\varepsilon, \alpha/2}(S_N), 1))$.

In the second part, we construct an $(\alpha/2, \delta)$-PAC prediction set for the threshold $\tau$ using $S'$ and a score function $g$ defined as $g(\tau, y) := \tau \mathbb{1}(y = 1)$, where $\mathbb{1}(\cdot)$ is the indicator function [2]. Thus, the constructed prediction set can be equivalently viewed as an interval $[\hat{\gamma}_{\alpha/2, \delta}(S'), \infty)$ where the thresholds $\tau$ of most calibration distributions are included; see Section 4.3 for a formal statement along with intuition on our design choice, and see Algorithm 1 for an implementation.

---

[1]It is also called a training-set conditional guarantee in [21].

[2]The constructed prediction set parameter is equivalent to $\tau$ satisfying $\mathbb{P}_S \left[ \tau \leq \hat{\gamma}_{\varepsilon, \alpha/2}(S) \right] \geq 1 - \alpha/2$ with probability at least $1 - \delta$ over the randomness of $S_1, \ldots, S_N$ used to estimate $\tau$.

**Adaptation and inference.** We then use labeled examples $A$ from a test distribution to adapt the meta-learned score function $f$ as in the meta calibration part. Then, we use the same parameter $\hat{\tau}(\boldsymbol{S}, \boldsymbol{A})$ of the meta prediction set along with the adapted score function as our final meta prediction set $F_{\hat{\tau}(\boldsymbol{S}, \boldsymbol{A})}$ for inference in the future.

Reusing the same meta prediction set parameter after adaptation is valid, as the meta prediction set parameter is also chosen after adaptation via a hold-out adaptation set from the same distribution as each respective calibration set. Moreover, by reusing the same parameter, we only need a few labeled examples from the test distribution for an adaptation set, without requiring additional labeled examples for calibration on the test distribution. Finally, we rely on a PAC prediction set algorithm that aims to minimize the expected prediction set size [21, 3]; Algorithm 1 relies on this property, thus making the meta prediction set size small.

### 4.3 Theory

The proposed Algorithm 1 outputs a threshold for a prediction set that satisfies the mPAC guarantee, as stated in the following result (see Appendix C for a proof):

**Theorem 1.** *The estimator $\hat{\tau}$ implemented in Algorithm 1 is $(\varepsilon, \alpha, \delta)$-mPAC for $p_\pi$.*

Intuitively, the algorithm implicitly constructs an interval $[\hat{\gamma}_{\alpha/2,\delta}(S'), \infty)$ over scalar parameters of prediction sets. In particular, for the $i$-th distribution, the Algorithm in Line 3 finds a scalar parameter of a prediction set for the $i$-th distribution, which forms an empirical distribution over the scalar parameters. Due to the i.i.d. nature of $\theta_i$, $i = 1, \ldots, N$, the parameter of a prediction set for a test distribution follows the same distribution over scalar parameters. Thus, choosing a conservative scalar parameter as in Line 5 of the Algorithm suffices to satisfy the mPAC guarantee.

## 5  Experiments

We demonstrate the efficacy of the proposed meta PAC prediction set based on four few-shot learning datasets across three application domains: mini-ImageNet [10] and CIFAR10-C [15] for the visual domain, FewRel [16] for the language domain, and CDC Heart Dataset [17] for the medical domain. See Appendix E for additional experiments, including the results for CIFAR10-C.

### 5.1  Experimental Setup

**Few-shot learning setup.** We consider $k$-shot $c$-way learning except for the CDC Heart Dataset; In particular, there are $c$ classes for each task dataset, and $k$ adaptation examples for each class. Thus, we have $t := kc$ labeled examples to adapt a model to a new task. As the CDC Heart dataset is a label unbalanced dataset, we do not assume equal shots per class.

**Score functions.** We use a prototypical network [11] as a score function. It uses $t$ adaptation examples to adapt the network to a given task dataset, and predicts labels on query examples.

**Baselines.** We compare four approaches, including our proposed approach.

- **PS**: We consider a naive application of PAC prediction sets [3]. We pool all $nN$ labeled examples from calibration datasets into one calibration set to run the PAC prediction set algorithm.
- **PS-Test**: We construct PAC prediction sets by using 20 shots for each class directly drawn from the test distribution.
- **Meta-CP**: We use the PAC variant of meta conformal prediction [14], which trains a quantile predictor along with a score function, while our approach only needs a score function. We run the authors' code with the same evaluation parameters for comparison.
- **Meta-PS**: This is the proposed meta PAC prediction set from Algorithm 1.

**Metric.** We evaluate prediction sets mainly via their empirical prediction set error over a test sample, *i.e.,* having learned a parameter $\tau$, we find $1/|E| \sum_{(x,y) \in E} \mathbb{1}(y \notin F_\tau^A(x))$, where $A$ is an adaptation set, and $E$ is an evaluation set drawn from each test task distribution. We desire this to be at most $\varepsilon$ given fixed calibration and test datasets. This should hold with probability at least $1 - \delta$ over the randomness due to the calibration and adaptation data; and with probability at least $1 - \alpha$ over the randomness due to the test adaptation set. To evaluate this, we conduct 100 random trials, drawing independent calibration datasets, and for each fixed calibration dataset, we conduct 50 random trials

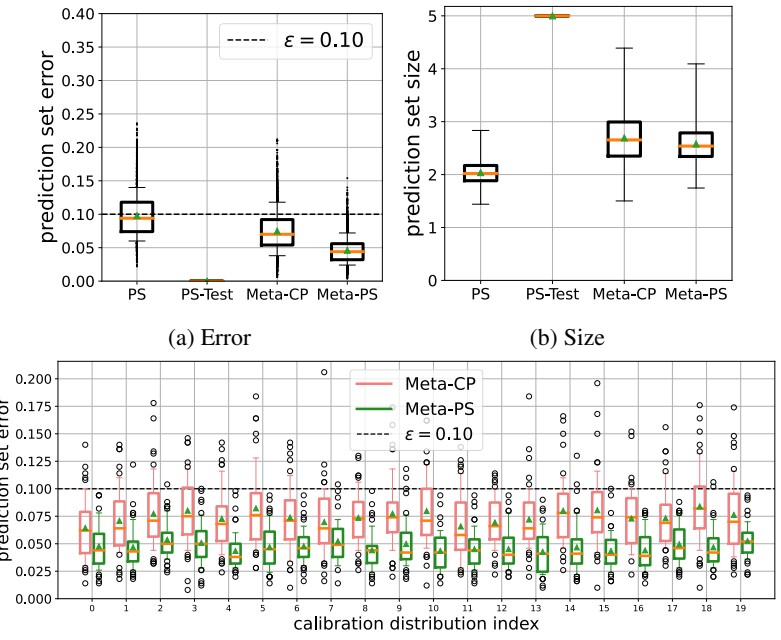

(a) Error

(b) Size

(c) Error given an augmented calibration set

Figure 3: Prediction set error (3a), size (3a), and error given a calibration set (3c) on mini-ImageNet. Parameters are $N = 500$, $n = 2500$, $t = 25$, $\varepsilon = 0.1$, $\alpha = 0.1$, $\delta = 10^{-5}$ for **Meta-PS** and $\varepsilon = 0.1$, $\delta = 10^{-5}$ for the other methods.

drawing independent test datasets. The prediction set size evaluation is done similarly to the error; we compute the empirical prediction set size $1/|E| \sum_{(x,y) \in E} S(F_\tau^A(x))$, where $S$ is a size measure, *e.g.,* set cardinality for classification.

## 5.2 Mini-ImageNet

Mini-ImageNet [10] is a smaller version of the ImageNet dataset for few-shot learning.

**Setup.** Mini-ImageNet consists of 100 classes with 64 classes for training, 16 classes for calibration, and 20 classes for testing; each class has 600 images. Following a standard setup, we consider five randomly chosen classes as one task and 5-way classification. In training, we follows 5-shot 5-way learning, considering $M = 800$ training task distribution randomly drawn from the $\binom{64}{5}$ possible tasks. In calibration, we have $N = 500$ calibration task datasets randomly drawn from the $\binom{16}{5}$ possible tasks, and use 5 shots for adaptation and 500 shots for calibration (*i.e., $t = 25$ and $n = 2500$*). In evaluation, we have 50 test task datasets randomly drawn from the $\binom{20}{5}$ possible tasks and use 5 shots from each of 5 classes for adaptation and 100 shots from each 5 classes for evaluation.

**Results.** Figure 3 shows the prediction set error and size of the various approaches in box plots. Whiskers in each box plot for the error cover the range between the 10-th and 90-th percentiles of the empirical error distribution, while whiskers in each box plot for the size represents the minimum and maximum of the empirical size distribution. The randomness of error and size is due to the randomness of the augmented calibration $(S, A)$ and a test adaptation set $A$; but since $\delta = 10^{-5}$, the randomness is mostly due to the adaptation set.

The prediction set error of the proposed approach is below the desired level $\varepsilon = 0.1$ at least a fraction $1 - \alpha = 0.9$ of the time. This supports that the proposed prediction set satisfies the meta PAC criterion. However, **PS** and **Meta-CP** do not empirically satisfy the meta PAC criterion, and **PS-Test** is too conservative. In particular, **Meta-CP** does not empirically control the randomness due to the test distribution. To make this point precise, Figure 3c shows a prediction set error distribution over different test distributions given a fixed calibration dataset. As can be seen, the proposed **Meta-PS** approach empirically satisfies the $\varepsilon$ error criterion at least a fraction $1 - \alpha = 0.9$ of the time (as

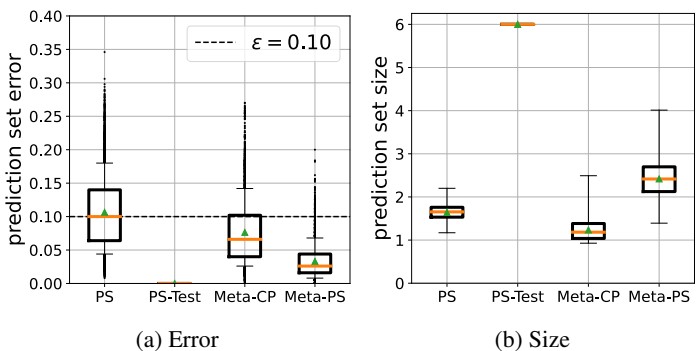

|  (a) Error | (b) Size |

Figure 4: Prediction set error and size on FewRel. Parameters are $N = 500$, $n = 2500$, $t = 25$, $\varepsilon = 0.1$, $\alpha = 0.1$, $\delta = 10^{-5}$ for **Meta-PS** and $\varepsilon = 0.1$, $\delta = 10^{-5}$ for other methods.

shown in the whiskers), but **Meta-CP** does not empirically control it. Figure 3b demonstrates that the proposed **Meta-PS** produces smaller prediction sets compared to the **PS-Test** approach that also satisfies the $\varepsilon$ error constraint.

### 5.3 FewRel

FewRel [16] is an entity relation dataset, where each example is the tuple of the first and second entities in a sentence along with their relation as a label.

**Setup.** FewRel consists of 100 classes, where 64 classes are used for training and 16 classes are originally used validation; each class has 700 examples. The detailed setup for training, calibration, and test are the same as those of mini-ImageNet, except that the test task distributions are randomly drawn from the original validation split.

**Results.** Figure 4 presents the prediction set error and size of the approaches. The trend of the results is similar to the mini-ImageNet result. Ours empirically satisfies the desired prediction set error specified by $\varepsilon$ with empirical probabilities of at least $1 - \alpha$ over the test adaptation set and $1 - \delta$ over the (augmented) calibration data. In contrast, the other approaches either do not empirically satisfy the meta PAC criterion or are highly conservative. Figure 6a also confirms that the proposed approach empirically controls the error with probability at least $1 - \alpha$ over the randomness in the test dataset.

### 5.4 CDC Heart Dataset

The CDC Behavioral Risk Factor Surveillance System [17] consists of health-related telephone surveys started in 1984. The survey data is collected from US residents. We use the data collected from 2011 to 2020 and consider predicting the risk of heart attack given other features (*e.g.,* level of exercise or weight). We call this curated dataset the CDC Heart Dataset. See Appendix D.4 for the details of the experiment setup.

**Setup.** The 10 years of the CDC Heart Dataset consists of 530 task distributions from various US states. We consider data from 2011-2014 as the training task distributions, consisting of 212 different tasks and 1,805,073 labeled examples, and data from 2015-2019 as calibration task distributions, consisting of 265 different tasks and 2,045,981 labeled examples. We use the data from 2020 as a test task distribution, consisting of 53 different tasks and 367,511 labeled examples. We use $M = 200$, $N = 250$, $n = 3000$, and $t = 10$; see Appendix D.4 for the details.

**Results.** Figure 5 includes the prediction set error and size of each approach. The trend is as before. The proposed **Meta-PS** empirically satisfies the $\varepsilon = 0.1$ constraint (*i.e.,* the top of the whisker is below of the $\varepsilon = 0.1$ line), while **PS** and **Meta-CP** violate this constraint. Meanwhile, **PS-Test** conservatively satisfies it. As before, Figure 6b empirically justifies that the proposed approach controls the prediction set error due to the randomness over the test adaptation data.

## 6 Conclusion

We propose a PAC prediction set algorithm for meta learning, which satisfies a PAC property tailored to a meta learning setup. The efficacy of the proposed algorithm is demonstrated in four datasets across three application domains. Specifically, we observe that the proposed algorithm finds a

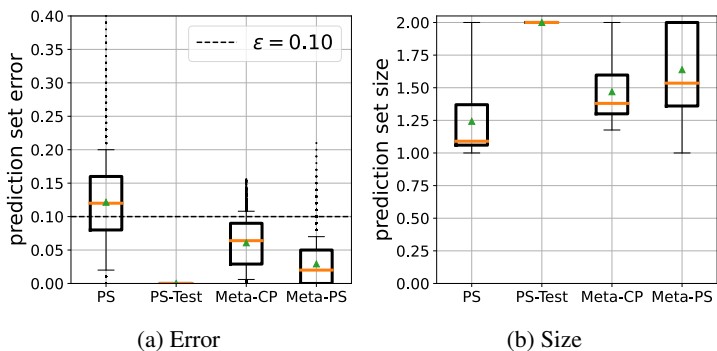

(a) Error

(b) Size

Figure 5: Prediction set error and size on CDC Heart Dataset. Parameters are $N = 250$, $n = 3000$, $t = 10$, $\varepsilon = 0.1$, $\alpha = 0.1$, $\delta = 10^{-5}$ for **Meta-PS** and $\varepsilon = 0.1$, $\delta = 10^{-5}$ for other methods.

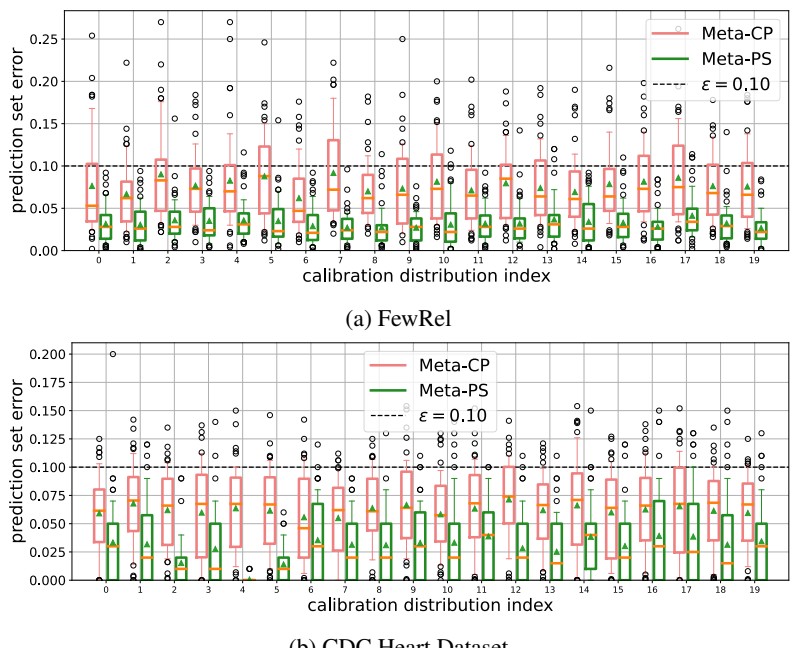

(a) FewRel

(b) CDC Heart Dataset

Figure 6: **Meta-PS** and **Meta-CP** prediction set error for various calibration sets. Given one calibration set (*i.e.,* a fixed calibration distribution index), the box plot with whiskers represents the $(\varepsilon, \alpha)$-guarantee. The proposed **Meta-PS** satisfies the $\varepsilon = 0.1$ constraints at least $1 - \alpha = 0.9$ of the time, as the whisker is below the dotted line indicating $\varepsilon$. However, **Meta-CP** does not satisfy this constraint, showing that the proposed approach can control the correctness on future tasks.

prediction set that satisfies a specified PAC guarantee, while producing a small set size. A limitation of the current approach is that it requires enough calibration datapoints to satisfy the PAC guarantee, see [3] for an analysis. A potential societal impact is that a user of the proposed algorithm might misuse the guarantee without a proper understanding of assumptions; see Appendix F for details.

## Acknowledgements

This work was supported in part by DARPA/AFRL FA8750-18-C-0090, ARO W911NF-20-1-0080, NSF TRIPODS 1934960, NSF DMS 2046874 (CAREER), NSF-Simons 2031895 (MoDL). Any opinions, findings and conclusions or recommendations expressed in this material are those of the authors and do not necessarily reflect the views of the Air Force Research Laboratory (AFRL), the Army Research Office (ARO), the Defense Advanced Research Projects Agency (DARPA), or the Department of Defense, or the United States Government.

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
