# A  Problem without Adaptation

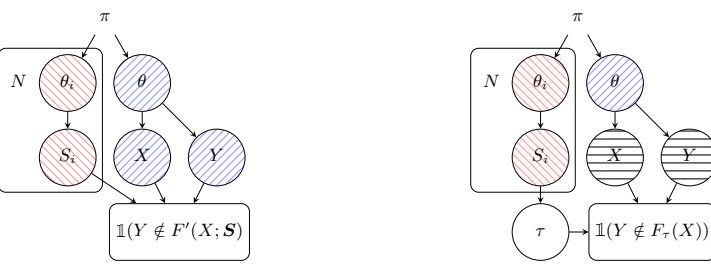

(a) Meta task-marginal guarantee w/o adaptation  (b) Meta PAC guarantee w/o adaptation (ours)

Figure 7: Comparison among different guarantees on prediction sets in no adaptation. An edge represents dependency between random variables, and a plate with $N$ represents that the variables in it are duplicated $N$ times. Each stripe pattern with a different color indicates that the corresponding random variables are simultaneously conditioned on in the correctness guarantee; $F'$ denotes a conformal prediction set. In the no adaptation setup, the proposed meta PAC prediction set (7b) has the most fine-grained control over the correctness guarantee conditioning over calibration tasks, the test task and adaptation set, and the test data.

We introduce our problem setup without adaptation, which is a special case of our main problem with adaptation, and thus may be simpler to understand.

**Setup.** In the setting of Section 3.2, the training samples from some training distributions $p_{\theta'_i}$, $i = 1, \ldots, M$ are used to learn a score function via an arbitrary meta learning method. We denote the given meta-learned score function by $f$. Additionally, we have $N$ calibration distributions $p_{\theta_1}, \ldots, p_{\theta_N}$, where $\theta_1, \ldots, \theta_N$ is an i.i.d. sample from $p_\pi$, *i.e.,* $\boldsymbol{\theta} := (\theta_1, \ldots, \theta_N) \sim p_\pi^N$. From each calibration distribution $p_{\theta_i}$, we draw $n$ i.i.d. labeled examples for a calibration set, *i.e.,* $\boldsymbol{S} := (S_1, \ldots, S_N) \sim p_{\theta_1}^n \cdots p_{\theta_N}^n$ and $\boldsymbol{S} \in \boldsymbol{\mathcal{S}} := \mathcal{Z}^{n \times N}$. Finally, we consider a test distribution $p_\theta$, where $\theta \sim p_\pi$, and draw i.i.d. labeled examples from it for an evaluation set.

**Problem.** Our goal is to find a prediction set $F_\tau : \mathcal{X} \to 2^{\mathcal{Y}}$ parametrized by $\tau \in \mathbb{R}_{\geq 0}$ which satisfies a meta-learning variant of the PAC definition. In particular, consider an estimator $\hat{\tau} : \boldsymbol{\mathcal{S}} \to \mathbb{R}_{\geq 0}$ that estimates the threshold $\tau$ defining a prediction set, given a calibration set $\boldsymbol{S}$.

Recall that a prediction set $F_\tau$ parametrized by $\tau$ is $\varepsilon$-correct for $p_\theta$ if $\tau \in T_\varepsilon(\theta)$. Moreover, we want the prediction set $F_\tau$, where $\tau$ is estimated via $\hat{\tau}_{\varepsilon,\alpha,\delta}$, to be correct on most test tasks $\theta$, *i.e.,* given $\boldsymbol{S} \in \boldsymbol{\mathcal{S}}$,

$$\mathbb{P}\left\{\hat{\tau}(\boldsymbol{S}) \in T_\varepsilon(\theta)\right\} \geq 1 - \alpha,$$

where the probability is taken over $\theta$. Intuitively, we want a prediction set to be correct on future labeled examples from $p_\theta$. Finally, we want the above to hold for most calibration sets $\boldsymbol{S}$ from most calibration tasks $\boldsymbol{\theta}$, leading to the following definition.

**Definition 4.** *An estimator $\hat{\tau} : \boldsymbol{\mathcal{S}} \to \mathbb{R}_{\geq 0}$ is $(\varepsilon, \alpha, \delta)$-meta probably approximately correct (mPAC) for $p_\pi$ if*

$$\mathbb{P}\left\{\mathbb{P}_\theta\left\{\hat{\tau}(\boldsymbol{S}) \in T_\varepsilon(\theta)\right\} \geq 1 - \alpha\right\} \geq 1 - \delta,$$

*where the outer probability is taken over $\boldsymbol{\theta}$ and $\boldsymbol{S}$ and the inner probability is taken over $\theta$.*

Figure 7b shows that the proposed correctness guarantee is conditioned over *calibration tasks*, a *test task*, and *test data* $(x, y)$. This implies that the correctness guarantee of a prediction set holds on most future test data from the same test task. In contrast, the guarantee in Figure 7a is conditioned on the entire test task.

# B  PAC Prediction Sets Algorithm

Algorithm 2 describes a way to find a PAC prediction set, proposed in [3, 6]. This is equivalent to inductive conformal prediction, tuned to achieve training-conditional validity [21]. Here, $\bar{\theta}(k; m, \delta)$

is the Clopper-Pearson upper bound [23] for the parameter of a Binomial distribution, where $F$ is the cumulative distribution function (CDF) of the binomial distribution $\mathrm{Binom}(m, \varepsilon)$ with $m$ trials and success probability $\varepsilon$, *i.e.,*

$$\overline{\theta}(k; m, \delta) := \inf \{\theta \in [0, 1] \mid F(k; m, \theta) \leq \delta\} \cup \{1\}.$$

---

**Algorithm 2** PAC Prediction Sets algorithm [3, 6].

---

   **procedure** PS-BINOM$(S, f, \varepsilon, \delta)$
      $\hat{\tau} \leftarrow 0$
      **for** $\tau \in \mathbb{R}_{\geq 0}$ **do**                                ($\triangleright$) Grid search in ascending order
         **if** $\overline{\theta}(\sum_{(x,y) \in S} \mathbb{1}(y \notin F_\tau(x)); |S|, \delta) \leq \varepsilon$ **then**
            $\hat{\tau} \leftarrow \max(\hat{\tau}, \tau)$
         **else**
            **break**
      **return** $\hat{\tau}$

---

## C   Proof of Theorem 1

We use the two defining properties of the estimators $\hat{\tau}$ and $\hat{\gamma}_{\varepsilon, \alpha/2}$; namely that $\hat{\tau}$ is $(\alpha/2, \delta)$-PAC and $\hat{\gamma}_{\varepsilon, \alpha/2}$ is $(\varepsilon, \alpha/2)$-PAC.

First, consider that the calibration set and adaptation set pair $(S_i, A_i)$ for $i = 1, \ldots, N$ can be viewed as an independent and identically distributed sample from the distributions $p_{\theta_i}^{n \times t}$ where $\theta_i \sim p_\pi$. Then, $\hat{\gamma}_{\varepsilon, \alpha/2}((S_i, A_i))$, where $A_i$ is used for adapting a score function, follows the distribution induced by $(S_i, A_i) \sim p_{\theta_i}^{n \times t}$. Since $[\hat{\tau}(\boldsymbol{S}, \boldsymbol{A}), \infty)$ is $(\alpha/2, \delta)$-PAC, we have

$$\mathbb{P}_{\boldsymbol{\theta}, \boldsymbol{S}, \boldsymbol{A}} \left\{ \mathbb{P}_{\theta, S, A} \left\{ \hat{\gamma}_{\varepsilon, \alpha/2}((S, A)) \geq \hat{\tau}(\boldsymbol{S}, \boldsymbol{A}) \right\} \geq 1 - \frac{\alpha}{2} \right\} \geq 1 - \delta. \tag{2}$$

Moreover, due to the $(\varepsilon, \alpha/2)$-PAC property of $\hat{\gamma}_{\varepsilon, \alpha/2}$, we have

$$\mathbb{P}_{\theta, S, A} \left\{ \hat{\gamma}_{\varepsilon, \alpha/2}((S, A)) \in T_\varepsilon(\theta, A) \right\} \geq 1 - \frac{\alpha}{2}.$$

It follows that

$$\mathbb{P}_{\theta, S, A} \left\{ \hat{\gamma}_{\varepsilon, \alpha/2}((S, A)) \leq \sup T_\varepsilon(\theta, A) \right\} \geq 1 - \frac{\alpha}{2}. \tag{3}$$

By a union bound, the events related to $(\theta, S, A)$ in (2) and (3) hold with probability at least $1 - \alpha$. Thus, we have

$$\mathbb{P}_{\boldsymbol{\theta}, \boldsymbol{S}, \boldsymbol{A}} \left\{ \mathbb{P}_{\theta, S, A} \left\{ \sup T_\varepsilon(\theta, A) \geq \hat{\tau}(\boldsymbol{S}, \boldsymbol{A}) \right\} \geq 1 - \alpha \right\} \geq 1 - \delta.$$

Since the inner expression does not depend of $S$, it follows that

$$\mathbb{P}_{\boldsymbol{\theta}, \boldsymbol{S}, \boldsymbol{A}} \left\{ \mathbb{P}_{\theta, A} \left\{ \sup T_\varepsilon(\theta, A) \geq \hat{\tau}(\boldsymbol{S}, \boldsymbol{A}) \right\} \geq 1 - \alpha \right\} \geq 1 - \delta.$$

Finally, from the definition of $\varepsilon$-correctness in (1) it follows directly that for any $\theta \in \Theta$ and any $\varepsilon \in [0, 1]$, if $\tau \in T_\varepsilon(\theta, A)$ and $0 \leq \tau' \leq \tau$, then $\tau' \in T_\varepsilon(\theta, A)$. Thus, it follows that

$$\mathbb{P}_{\boldsymbol{\theta}, \boldsymbol{S}, \boldsymbol{A}} \left\{ \mathbb{P}_{\theta, A} \left\{ \hat{\tau}(\boldsymbol{S}, \boldsymbol{A}) \in T_\varepsilon(\theta, A) \right\} \geq 1 - \alpha \right\} \geq 1 - \delta,$$

as claimed.

## D   Experiment Details

### D.1   Computing Environment

The experiments are done using an NVIDIA RTX A6000 GPU and an AMD EPYC 7402 24-Core CPU.

### D.2 Dataset License, Consent, and Privacy Issues

mini-ImageNet [10] is licensed under the the MIT License [3], CIFAR10-C [15] is licensed under the the Apache 2.0 License [4], and FewRel [16] is licensed under the MIT License [5]. The CDC Heart Dataset [17] is produced by the Centers for Disease Control and Prevention, a US federal agency, and belongs to the public domain.

mini-ImageNet, CIFAR10-C, and FewRel use public data (*e.g.,* images from Web or corpuses from Wikipedia), so generally consent is not required. The CDC Heart Dataset is based on a survey, which is collected in-person from consenting individuals.

mini-ImageNet, CIFAR10-C, and FewRel rarely contain personally identifiable information or offensive content, as verified over several years. For the CDC Heart Dataset, personally identifiable information is excluded from the publically available data.

### D.3 Prototypical Network Training

We train a prototypical network using an Adam optimizer with a dataset-specific initial learning rate, decaying it by a factor of two for every 40 training epochs. We use an initial learning rate of 0.001 for mini-ImageNet, 0.01 for CIFAR10-C, 0.01 for FewRel, and 0.001 for the CDC Heart Dataset.

For the backbone of the prototypical network, we use a four-layer convolutional neural network (CNN) for mini-ImageNet as in [11, 14], a ResNet-50 for CIFAR10-C, a CNN sentence encoder for FewRel as in [16, 14], and a two-layer fully connected network for CDC Heart Dataset, where each layer consists of 100 neurons followed by the ReLU activation layer and the Dropout layer with a dropout probability of 0.5.

### D.4 Details of CDC Heart Dataset Experiment

**Data Post-processing.** We curate the original data released by the CDC [17]; in particular, the original data has a variable number of features per year (varying from 275 to 454), where each feature corresponds to a survey item. Among these features, we use the answer of "Ever Diagnosed with Heart Attack" as our label, which corresponds to the feature with SAS variable `CVDINFR4`. We use features reported in each year. Then, we remove certain non-informative features (date, year, month, day, and sequence number) and features with more than 5% of values missing. Furthermore, each datapoint that contains any missing value or has a missing label is also removed, resulting in our final dataset used in our experiments.

**Prototypical Network Modification.** The CDC Heart Dataset is a highly unbalanced dataset, as the ratio of the positive labels is about 6%. This does not fit with the conventional label-balance assumption of few-shot learning [11]. To handle the unbalanced labels, we modify prototypical networks to generate prototypes only for the labels observed in the adaptation set, while producing zero prediction probabilities for unobserved labels.

## E  Additional Experiments

### E.1 Corrupted CIFAR10 (CIFAR10-C)

CIFAR10-C is a CIFAR10 dataset with synthetic corruptions. In particular, the corruptions consist of 15 synthetic image perturbations (*e.g.,* Gaussian noise) with 6 different severity levels, ranging from 0 to 5.

**Setup.** To use CIFAR10-C as a few-shot learning dataset, we let a set of corruptions define one task dataset. In particular, we choose all stochastic corruptions from 15 perturbations (*i.e.,* Gaussian noise, shot noise, impulse noise, and elastic transform) along with three severity levels (namely, 0, 1, and 2). Thus, we can obtain nine different corruptions. Then, a subsequence of the nine corruptions of size at most three is randomly chosen and applied to CIFAR10 to define one task distribution. Moreover, we use CIFAR10 labels, leading to 10-way learning.

---

[3] https://github.com/yaoyao-liu/mini-imagenet-tools/blob/main/LICENSE
[4] https://github.com/hendrycks/robustness/blob/master/LICENSE
[5] https://github.com/thunlp/FewRel/blob/master/LICENSE

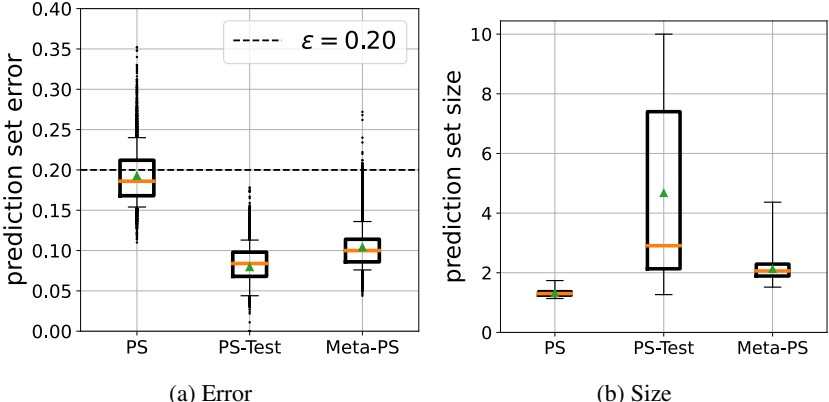

(a) Error                  (b) Size

Figure 8: Prediction set error and size on CIFAR10-C. Parameters are $N = 500$, $n = 5000$, $t = 50$, $\varepsilon = 0.2$, $\alpha = 0.1$, $\delta = 10^{-5}$ for **Meta-PS** and $\varepsilon = 0.2$, $\delta = 10^{-5}$ for other methods.

In training, we consider $M = 1000$ training task distributions and train a score function via 5-shot 10-way learning. During calibration, we consider $N = 500$ calibration task distributions and 500 calibration shots for each class (*i.e.,* $n = 5000$) and 5 adaptation shots for each class (*i.e.,* $t = 50$). During testing, we have 50 test task distributions and use 5 shots per class for adaptation and 100 shots per class for evaluation.

**Results.** Figure 8 demonstrates the efficacy of the proposed approach. **Meta-PS** satisfies PAC constraint with $\varepsilon = 0.2$ a fraction $1 - \alpha = 0.9$ of the time (as shown in whiskers). The randomness of an augmented calibration set is negligible, as we set $\delta = 10^{-5}$. Other approaches either violate the $\varepsilon$ constraint or are more conservative than the proposed approach. In particular, **PS-Test** is slightly worse than the proposed approach. Considering that it requires 200 additional labeled examples along with 50 adaptation examples from a test distribution for calibration, **PS-Test** is more expansive than **Meta-PS**, which only needs 50 adaptation examples during testing.

### E.2 Prediction Set Error Per Calibration Set for Varying $\delta$

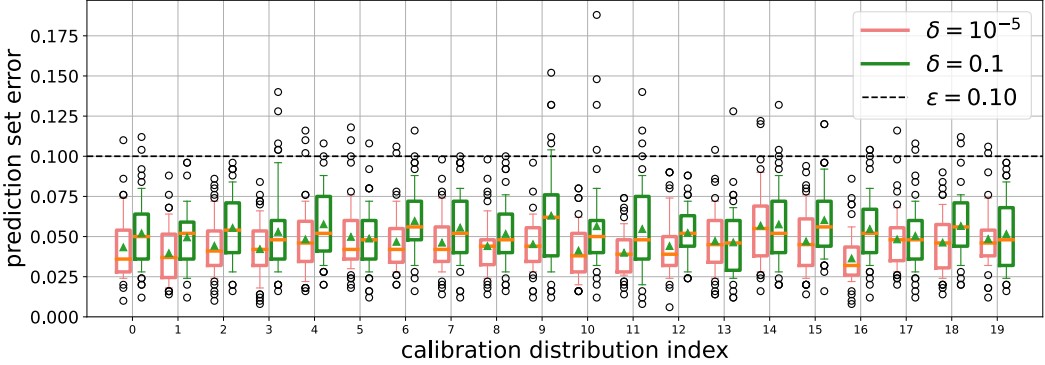

Figure 9: **Meta-PS** prediction set error for various calibration sets on different $\delta$ over Mini-ImageNet. Here $\delta = 0.1$ means at most $10\%$ of the calibration sets can produce prediction sets that violate the $\varepsilon$-constraint with probability at most $\alpha = 0.1$. This means the green whisker can be over the line showing $\varepsilon = 0.1$, as for the calibration distribution with index 9. Based on this interpretation, the orange whisker is sufficiently below of the $\varepsilon = 0.1$ line, as $\delta = 10^{-5}$.

## F    Discussion

**On the i.i.d. assumption over tasks.** We believe that the i.i.d. assumption is useful and even reasonable in many settings. For instance, the most common meta-learning setting is where users

want to add new labels to an existing system, and we believe the arrival of new labels can reasonably be modeled as an i.i.d. process. This means that labels are thought of as belonging to the collection of "all possible labels", and are revealed without any systematic pattern or connection among themselves. Of course, this assumption does not need to be exactly satisfied, but the same is true of i.i.d. assumptions on the data that are common to prove any guarantees in learning theory. Alleviating this assumption is an important direction for future work.

**On the necessity of assumptions.** The proposed algorithm under assumptions (*e.g.,* the i.i.d. assumption on tasks) may not be a cure-all for meta-learning calibration. However, we believe that having a rigorous guarantee—under appropriate assumptions—significantly increases its importance.