# OpenReview forum: "PAC Prediction Sets for Meta-Learning"
_NeurIPS.cc/2022/Conference — NeurIPS 2022 Accept_

### Official Review · Reviewer_s7jQ · 2022-07-03

**Rating:** 6
**Confidence:** 3
**Soundness:** 3 good
**Presentation:** 3 good
**Contribution:** 3 good

**Summary:**

This paper proposes a PAC prediction set algorithm for meta learning. The goal is to adapt the predictor to new unseen tasks with a few labeled training examples. Experiments on four datasets demonstrate the effectiveness of the proposed methods with  PAC guarantee while having smaller prediction set size.


**Questions:**

Please refer to weaknesss above.

**Limitations:**

Yes, the authors have addressed the limitations and potential negative societal impact of their work.

**Strengths And Weaknesses:**

Strengths:


* The paper is well-written.
* The proposed method is reasonable with theoretical analysis.
* Experiments on various domain and datasets are compared.


Weakness:



* For reusing the same meta prediction set parameter after adaptation, would be better if the meta predictions set parameters are also adapted?
* The current methods seem only applies on metric-based meta-learning methods, e.g., prototypical network. It would be better if the methods could be also applies on optimization-based methods, such as model-agnostic meta learning (MAML).
* For mini-ImageNet datasets, only 5-way classification problem is compared. It would be better if the authors can compare on more challenging, 10-way or 20-way classification.

---

> ### Author Response · Authors · 2022-08-01
> **Response**
>
> We are thankful for the reviewer’s interest and comments; we will share the revised paper soon.
>
> ---
>
> > Weakness 1. For reusing the same meta prediction set parameter after adaptation, would be better if the meta predictions set parameters are also adapted?
>
> It is difficult to adapt the meta prediction set threshold because we have very few samples for each domain, making it essentially impossible to statistically quantify the quality of the scoring function in that domain. As a consequence, we focus on learning a single, fixed threshold, and relying on using a meta-learning algorithm to effectively adapt the scoring function to the new domain using a few samples.
>
> ---
>
> > Weakness 2. The current methods seem only applies on metric-based meta-learning methods, e.g., prototypical network. It would be better if the methods could be also applies on optimization-based methods, such as model-agnostic meta learning (MAML).
>
> We emphasize that *any* meta-learning algorithm can be used to adapt the scoring function to the new domain, including MAML. Our approach provides guarantees regardless of how the scoring function is adapted, as long as the same algorithm is used for all tasks.
>
> ---
>
> > Weakness 3. For mini-ImageNet datasets, only 5-way classification problem is compared. It would be better if the authors can compare on more challenging, 10-way or 20-way classification.
>
> We choose 5-shot 5-way classification as it is one of the dominant setups in meta learning papers; our approach could easily be applied to 10-way or 20-way settings. The main bottleneck is the quality of the scoring function, so as long as an effective meta-learning algorithm is available for training a good scoring function, then our approach should work well.

---

### Official Review · Reviewer_MYhn · 2022-07-07

**Rating:** 5
**Confidence:** 2
**Soundness:** 2 fair
**Presentation:** 2 fair
**Contribution:** 2 fair

**Summary:**

The paper proposed a method for estimating the prediction sets in the meta-test stage from the estimated prediction sets gathered from the meta-train adaptation and calibration phases. The whole process is carefully designed to satisfy the PAC setting in provide a theoretical guarantee for generating the meta-test prediction sets without calibration. The proposed method is tested on four different datasets across three application domains with competitive results.

**Questions:**

1. In terms of evaluation, I think both predictions set error and prediction set size are considered as the measurements. Among the experiment settings, Meta-PS shows relatively low set error but in some cases set size is larger than others. I think the idea case is that both prediction set error and prediction set size are small. So can the author draw a clear conclusion for the comparison between Meta-CP and Meta-PS?
2. The authors select PS-Binom to estimate \tau. Do different estimators make difference in the performance?
3. The score functions in lines 108 and 106 have different input spaces. Can the author explain why? My understanding is that f is a meta-learned off-the-shelf model.

**Limitations:**

1. The paper is not self-contain, the readers who are not familiar with [1] will have a hard time understanding this paper. I will highly suggest the authors to spent some paragraphs explaining some background.
2. Only one meta-model, prototypical network, is applied as a score function to evaluate the proposed method.
3. The notations are very confusing, especially those applied for the score function, and prediction set.
[1] Park, Sangdon, et al. "PAC confidence sets for deep neural networks via calibrated prediction." arXiv preprint arXiv:2001.00106 (2019).

**Strengths And Weaknesses:**

Strengths:
1. The proposed method is guaranteed by PAC bound.

Weaknesses:
1. The paper is not presented in a good way. The notation is very confusing and misleading in general.
2. A big picture is missing. For example, we know that f is a pre-trained meta-model. But in Figure 1 this information is missing which may confuse the meta-learning people.

---

> ### Author Response · Authors · 2022-08-01
> **Response**
>
> Thanks for valuable feedback! We will share the revised paper soon.
>
> ---
>
> > Weakness 1. The paper is not presented in a good way. The notation is very confusing and misleading in general.
>
> Thank you for pointing this out; we added the following to enhance the clarity:
> * We added details on Figure 2.
> * We used F_\tau for the notation of a prediction set instead of f_\tau
> * We added a brief overview in plain words before the problem formulation.
>
> ---
>
> > Weakness 2. A big picture is missing. For example, we know that f is a pre-trained meta-model. But in Figure 1 this information is missing which may confuse the meta-learning people.
>
> Yes, $f$ is a pretrained model, which we assume is given. Importantly, we note that $f$ may be adapted to new domains using any meta learning algorithm. We have added this information in the caption.
>
> ---
>
> > Question 1. In terms of evaluation, I think both predictions set error and prediction set size are considered as the measurements. Among the experiment settings, Meta-PS shows relatively low set error but in some cases set size is larger than others. I think the idea case is that both prediction set error and prediction set size are small. So can the author draw a clear conclusion for the comparison between Meta-CP and Meta-PS?
>
> Due to the scalar parameterization of prediction sets in the prediction set literature, there is a tradeoff between prediction set error and prediction set size. Thus, it is impossible to obtain both strictly smaller error and strictly smaller size.
>
> Thus, the goal is to minimize prediction set size subject to a constraint that the desired prediction set error bound is satisfied. Thus, our approach performs best when it obtains the smallest prediction set size among baselines that satisfy the error bound (assuming our approach itself satisfies the error bound).
>
> In this sense, our algorithm satisfies the meta-PAC guarantee (i.e., the dotted epsilon error line), but Meta-CP does not. Note that this behavior of Meta-CP is expected as it satisfies another guarantee; in general, choosing a right guarantee depends on applications, but we believe our guarantee is highly desirable since it controls the error across new tasks, whereas Meta-CP does not.
>
> We have added a more detailed explanation of the error and size tradeoff to Section 3.1.
>
> ---
>
> > Question 2. The authors select PS-Binom to estimate \tau. Do different estimators make difference in the performance?
>
> As long as the different estimator satisfies the required PAC guarantee, there is no difference in terms of satisfying the meta PAC guarantee. However, the prediction set size can vary depending on the statistical efficiency of the estimators, which is our secondary performance criterion.
>
> ---
>
> > Question 3. The score functions in lines 108 and 106 have different input spaces. Can the author explain why? My understanding is that f is a meta-learned off-the-shelf model.
>
> The score function in line 106 is a special case of the score function in line 108 when there are no adaptation samples (i.e., $t=0$). As mentioned, $f$ is a meta-learned off-the-shelf model, which can be either via zero-shot learning and few-shot learning. Here, we want to highlight that our approach works for both meta-learning setups, but to avoid confusion, we updated to use a different notation on the score function when there is no adaptation.
>
> ---
>
> > Limitation 1. The paper is not self-contain, the readers who are not familiar with [1] will have a hard time understanding this paper. I will highly suggest the authors to spent some paragraphs explaining some background.
>
> Section 3.1 is intended to be a background summary of [1], but we have added more details (e.g., the relationship between prediction set size and error).
>
> ---
>
> > Limitation 2. Only one meta-model, prototypical network, is applied as a score function to evaluate the proposed method.
>
> The guarantee holds for *any* score function; this is why we choose a meta-model, also followed by Meta-CP [14]. Different meta-models can change the prediction set size, but our paper is focused on having the meta-PAC guarantee, rather than reducing the set size.
>
> ---
>
> > Limitation 3. The notations are very confusing, especially those applied for the score function, and prediction set.
>
> We revised the paper to use different notations for score functions and prediction sets (i.e., f for score functions and F for prediction sets).

---

> > ### Comment · Reviewer_MYhn · 2022-08-09
> > **Response**
> >
> > Since the author addressed my concern, I rise my score. But I am not really an expert in this field.

---

> > > ### Author Response · Authors · 2022-08-10
> > > **Thanks!**
> > >
> > > Thanks for your response and positive evaluation!

---

### Official Review · Reviewer_e4fH · 2022-07-11

**Rating:** 6
**Confidence:** 5
**Soundness:** 4 excellent
**Presentation:** 2 fair
**Contribution:** 2 fair

**Summary:**

POST DISCUSSION:

The authors addressed most of my issues about readability and the attribution problem, so I have increased my score to a weak accept.

My final assessment is as follows:
I believe the proofs are correct, and the related work is correct.
I think the paper is the first to do a two-stage training-conditional guarantee, which has some value. Although this guarantee is a straightforward extension of existing results, perhaps somebody might be interested in it, and I do not believe it has appeared before.

I also believe it contributes by being the second paper to use training-conditional conformal for meta-learning. It is slightly different from the existing work by Adam Fisch et al. in that it also conditions on the tasks. Maybe somebody will find that useful!

_____________
This paper introduces a form of distribution-free uncertainty quantification guarantee into meta-learning and proposes an algorithm to achieve it. The guarantee is a triply-nested one, where the innermost integral is an expectation on a 0-1 loss. This expectation is nested within two other probabilities, conditioned on the calibration set and the meta learned parameters.  The algorithm boils down to two applications of binomial concentration. There are then extensive experiments on several meta learning tasks in vision, language, and medicine.

**Questions:**

What does it mean for tasks to be “iid” in the real world, outside of synthetic setups? What are some realistic settings where we might expect the guarantee provided from the authors to actually hold?

**Limitations:**

I believe the limitations have been addressed. (I do not believe there are any serious ethical limitations of this paper in the first place, and ethics review should certainly not get in the way of the publication of this paper.)

**Strengths And Weaknesses:**

Strengths: I the technical results are clearly correct. The experiments are comprehensive.

Weaknesses:

(1) The guarantee provided in this paper is so complicated to interpret that I worry it will never be used.  It is already difficult enough to interpret a tolerance interval, and adding an extra parameter incurs such a high cognitive cost for the user that one may never know how to set these parameters.

(2) The paper is difficult to understand, as the writing is dense and exceedingly complicated. For example, take Figure 2. It includes a lot of extraneous information, the colors are missing, and certain aspects of the plot are not explained. What do the orientations of the stripes mean, for example? What does the box with “N” mean? Another part that is hard to read is 4.1.  There is just too much going on and I’m not sure anybody except for a very small community working on these PAC type guarantees will understand it.

(3) The technical result is not too interesting and the domain area is not novel.  The meta conformal prediction paper already provides a guarantee in this problem domain that works well, although it is marginal on the meta learned parameters. The fact that we can apply binomial concentration twice as opposed to once in order to condition again on those variables is basically an extension of existing results. Having said that, I think it is interesting and clever that we can chain the concentration argument twice. Nice work on the algorithm.

(4) I believe there is a major attribution issue throughout the work. The idea of training-conditional prediction sets was first proposed by Vovk [21]. Rebranding it as PAC prediction sets and citing both works as the source of the idea is wrong. The idea comes from Vovk and his paper should get all of the credit for that.  Please remove all instances of the following attribution: “PAC prediction sets [3]”. The same is true of the so-called PS-Binom algorithm, which is essentially the same as that of Vovk in Proposition 2b of [21].  The paper [3] can be credited for applying [21] in a deep learning setting, but [21] should be cited as the sole inventor of this idea, in this and any future work. I also take some issue with the rebranding as PAC prediction sets as opposed to training-conditional validity, as they are exactly equivalent and the latter terminology came first. I will argue for rejection of the paper unless these attributions are amended.

---

> ### Author Response · Authors · 2022-08-01
> **Response (1/2)**
>
> We appreciate considerable comments; we will share the revised paper soon.
>
> > (1) The guarantee provided in this paper is so complicated to interpret that I worry it will never be used. It is already difficult enough to interpret a tolerance interval, and adding an extra parameter incurs such a high cognitive cost for the user that one may never know how to set these parameters.
>
> We disagree that the guarantee is complicated to interpret; basically, it says that we want to (i) ensure high coverage (the epsilon), (ii) with high probability over the training data (the delta), and (iii) for most future tasks (the alpha). We will do our best to convey this intuition in the paper. Prior work has studied (i) and (ii), but not (iii), which is important when each instance of a task should satisfy the guarantee with high probability. Our approach is designed to provide this guarantee.
>
> For choosing the parameters, all three parameters are user-defined specifications, which can be any desired values for the correctness guarantee. One simple guideline is to choose $\epsilon$ and $\alpha$ equal (i.e., $epsilon=alpha$) based on the desired coverage rate, and choosing $\delta$ to be very small (e.g., $\delta=10^{-3}$), since the approach scales well to small $\delta$. This strategy ensures the desired coverage across most tasks.
>
> ---
>
> > (2) The paper is difficult to understand, as the writing is dense and exceedingly complicated. For example, take Figure 2. It includes a lot of extraneous information, the colors are missing, and certain aspects of the plot are not explained. What do the orientations of the stripes mean, for example? What does the box with “N” mean? Another part that is hard to read is 4.1. There is just too much going on and I’m not sure anybody except for a very small community working on these PAC type guarantees will understand it.
>
> We will upload a revised paper with highlight on updated part; for a short explanation,
> * In Figure 2, we will move (c-d) to the appendix, while adding details on extraneous information.
> * In Figure 2, we use three colors (green, red, and blue) with different patterns, each of which means a group of random variables conditioned at the same time.
> * We will explain the meaning of the graph (e.g., the meaning of edges, and ‘N’, which is the number of plates in drawing a graphical model)
> * We have a simplified version of 4.1. in Appendix A, but we also added a plain explanation before 4.1.
>
> ---
>
> > (3) The technical result is not too interesting and the domain area is not novel. The meta conformal prediction paper already provides a guarantee in this problem domain that works well, although it is marginal on the meta learned parameters. The fact that we can apply binomial concentration twice as opposed to once in order to condition again on those variables is basically an extension of existing results. Having said that, I think it is interesting and clever that we can chain the concentration argument twice. Nice work on the algorithm.
>
> We believe that conditioning on the task is an important condition for many safety-critical settings. For example, consider a robot using meta-learning to adapt to new environments. It is not sufficient to ensure that the robot works well on average across environments in which it operates, since it might fail to satisfy the coverage guarantee in half the environments. Instead, we want to ensure it works with high probability in each new environment.
>
> Further, we think the domain area of meta-learning is very important and of course, widely studied. Our work provides theoretical guarantees for prediction sets in meta-learning, which is a new area with only a handful of papers.
>
> We think that the analysis has novel elements, which have not appeared in the literature before. Also, we do not think it is fair to criticize our work as “basically an extension”, because in fact the same criticism would apply to all previous papers in the area (including the previous meta-learning paper by Fisch et al); and as we mentioned our paper has novel theoretical ideas.

---

> > ### Comment · Reviewer_e4fH · 2022-08-03
> > **Thank you for the responses**
> >
> > "We disagree that the guarantee is complicated to interpret; basically, it says that we want to (i) ensure high coverage (the epsilon), (ii) with high probability over the training data (the delta), and (iii) for most future tasks (the alpha)."
> >
> > When you explain it like that, I think somebody might be able to understand the guarantee.  I'd encourage the authors to _significantly_ simplify the notation of the main theorem, and to rewrite the text to get this simple point across. Right now, the theorem is a thicket of notation. I'm just trying to help and be honest with you --- only a small fraction of your potential audience readers will understand the theory section as currently written. There are too many symbols with unclear and possibly unnecessary subscripts and superscripts. As evidence towards this fact, Reviewer TeH4 and Reviewer MYhn said the notation was burdensome and confusing.
> >
> > Looking forward to seeing the revised paper. In addition to the simplifications you described, I really think it would go a long way to simplify the mathematical notation.
> >
> > "We think that the analysis has novel elements, which have not appeared in the literature before. Also, we do not think it is fair to criticize our work as “basically an extension”, because in fact the same criticism would apply to all previous papers in the area (including the previous meta-learning paper by Fisch et al); and as we mentioned our paper has novel theoretical ideas."
> >
> > I would also describe the paper by Fisch et al. as an application of existing algorithms, but the reason that paper gets a bit more credit is because they were the first to apply conformal ideas to the meta learning setting. I don't think Fisch et al. or this paper bring in new theoretical ideas (i.e., contributions to the statistical methodology of conformal prediction). Not every paper has a statistical contribution, and that's ok.
> >
> > _________________________________________________
> >
> > Just to be clear, the primary reason why my score is a "reject" is that I don't feel the paper is understandable as-written, and thus it won't be useful to the intended audience. The secondary reason is the attribution issue, which I hope we can jointly resolve.
> >
> > The lack of statistical novelty is not the reason my score is a "reject"; I'm just pointing it out because it's my job as a reviewer to assess novelty.

---

> > > ### Author Response · Authors · 2022-08-05
> > > **Thank you for the clarification**
> > >
> > > Thanks for the clarification! We have simplified notations and the revised paper is just updated. Let us know if you have any concerns.

---

> > > > ### Comment · Reviewer_e4fH · 2022-08-07
> > > > **I think the new writing is a lot better**
> > > >
> > > > Lines 133-144 help a lot, thanks for adding them.
> > > > I actually think they could be condensed, or at least summarized. Something like the version you have in this comment might be even better: "(i) ensure high coverage (the epsilon), (ii) with high probability over the training data (the delta), and (iii) for most future tasks (the alpha)."
> > > >
> > > > It also might help the clarity to have something with underbraces like this:
> > > >
> > > > $\mathbb{P}\Big\{ \mathbb{P}\big\{  \hat{\tau} \in \underbrace{T_{\epsilon}(\theta, A)}_{1-\epsilon\text{ coverage}} \big\} \geq \underbrace{1-\alpha}_{\text{for most training sets}} \Big\} \geq \underbrace{1-\delta}_{\text{for most tasks}}$
> > > >
> > > > (It doesn't render properly for me in OpenReview but you can paste it here to see what I mean: https://quicklatex.com)
> > > >
> > > > Anyway, these are just suggestions.
> > > >
> > > > In my mind, this issue is mostly resolved, and the writing is much better.  If we also finish handling the attribution problem, I will change my score to a weak accept (a large upgrade of 3 points). At the moment, I still feel there are errors there.

---

> ### Author Response · Authors · 2022-08-01
> **Response (2/2)**
>
> > (4) I believe there is a major attribution issue throughout the work. The idea of training-conditional prediction sets was first proposed by Vovk [21]. Rebranding it as PAC prediction sets and citing both works as the source of the idea is wrong. The idea comes from Vovk and his paper should get all of the credit for that. Please remove all instances of the following attribution: “PAC prediction sets [3]”. The same is true of the so-called PS-Binom algorithm, which is essentially the same as that of Vovk in Proposition 2b of [21]. The paper [3] can be credited for applying [21] in a deep learning setting, but [21] should be cited as the sole inventor of this idea, in this and any future work. I also take some issue with the rebranding as PAC prediction sets as opposed to training-conditional validity, as they are exactly equivalent and the latter terminology came first. I will argue for rejection of the paper unless these attributions are amended.
>
> We omitted a more careful discussion of attribution due to space concerns, and will expand it in our revision --- our intent was to attribute the original idea to Vovk [21] and Wilks [1], while the perspective of PAC prediction sets as a learning problem is due to [3]. Specifically, as pointed out in Vovk [21], the basic idea of label coverage without covariates was due to Wilks (1941) [1] in his work on tolerance regions. Vovk’s 2013 paper extended this idea to the conformal prediction setting, with training-set conditional guarantees marginally over covariates $x$, basically by applying Wilks’ algorithm to the non-conformity scores $f(x,y)$. The formulation in [3] is mathematically equivalent, but views the estimation of the threshold $\tau$ as a learning problem, such that obtaining a coverage guarantee is equivalent to proving a generalization bound. This leads to a more efficient inference algorithm (i.e., O(1) rather than O(n) for [21] with respect to the number of calibration samples n due to scalar parameterization). We build on this perspective in our paper since it helps inform how to extend the strategy to the meta-learning setting. As one can clearly see from our algorithm, we apply a prediction set for the scalars $\tau$ at one stage, which is only possible due to the perspective from [3], which takes $\tau$ as the key object of study. Therefore, our algorithm directly builds on the prior work from [3], and would be more cumbersome in terms of the analysis in [21].  We will elaborate on the specific innovations of each prior work in our paper and correct the attribution, and we are happy to make additional modifications to the above discussion if there are concerns.
>
> On naming, we adopted the term “PAC prediction sets” from [6] due to the following reasons.
> First, we note that we would have to use the term “training-conditional valid prediction sets”, since “training-conditional” refers to the validity guarantee. We believe “PAC” is more well-known in the machine learning community (also, “PAC” predates “training-conditional”, though not in the context of conformal prediction).
>
> ---
>
> > Questions: What does it mean for tasks to be “iid” in the real world, outside of synthetic setups? What are some realistic settings where we might expect the guarantee provided from the authors to actually hold?
>
> We believe that the iid assumption is useful and even reasonable in many settings. For instance, the most common meta-learning setting is where users want to add new labels to an existing system, and we believe the arrival of new labels can reasonably be modeled as an iid process. Of course, this assumption does not need to be exactly satisfied, but the same is true of iid assumptions on the data that are common to prove any guarantees in learning theory.  Alleviating this assumption is an important direction for future work. We will add a discussion to our paper.

---

> > ### Comment · Reviewer_e4fH · 2022-08-03
> > **Can you clarify?**
> >
> > I am not sure if I agree with several points the authors make, and would appreciate clarification.
> >
> > Mainly, I believe the PS-Binom algorithm in [3] is actually exactly equivalent to inverting (7) in [21], and solving for $\epsilon$ (which is exactly $\tau$ in the PS-Binom algorithm). I believe this can be done trivially, with only a couple of calls to standard functions such as the inverse CDF of a Binomial. If this is true, then [3] did not contribute a new algorithm, and in fact PS-Binom may be suboptimal because it requires searching over a discretized grid while I believe that the inversion of (7) can be done analytically. In particular, in the setting of (7), I believe you can take
> >
> > $\epsilon = \frac{\mathrm{BinoInv}_{n,E}(\delta)+2-\rho}{n+1}$,
> >
> > where $\mathrm{BinoInv}_{n,E}$ is the inverse Binomial CDF with $n$ samples and success probability $E$, and $\rho$ is some tiny constant approaching $0$ (in practice you can take $\rho=1e-100$).
> > This should solve for the largest $\epsilon$ that gives an $(E,\delta)-valid$ prediction set.
> > The parameter $\epsilon$ is analogous to $\tau$ in the PS-Binom algorithm, and the procedure I just described above is significantly more efficient than the PS-Binom algorithm as it does not require any computation at all. I believe it is also exactly tight and finds the same solution as the PS-Binom algorithm (up to discretization error in the selection of the PS-Binom grid or equivalently the parameter $\rho$).
> >
> > I do not understand how the authors have interpreted this algorithm from [21] as taking $\mathcal{O}(n)$ computations. It clearly does not scale in complexity with the sample size and can be calculated trivially.
> >
> > Do the authors agree?

---

> > > ### Author Response · Authors · 2022-08-05
> > > **Clarificaiton**
> > >
> > > Thanks for the quick response! The computational time we are referring to is for constructing a prediction set for a new input x, *not* the computation time for estimating $\tau$ for [3] or $\epsilon$ for [21]. Also, we note that $\tau$ and $\epsilon$ play very different roles: $\tau$ is a threshold on the scoring function, whereas $\epsilon$ is an error rate (more closely related to the number of errors in PS-Binom, i.e., $\sum_{(x, y) \in S} 1( y \notin F_\tau(x))$). In particular, for prediction sets parameterized by a scalar value (e.g., [3]), the prediction set is represented as follows:
> > >
> > > $
> > > F_\tau (x) = \\{ y \in \mathcal{Y} \mid f(x, y) \ge \tau  \\}.
> > > $
> > >
> > > Given $x$ and estimated $\tau$, we only need to compute $f(x, y) \ge \tau$ for each y, which does not depend on the number of calibration examples, thus $O(1)$.
> > >
> > > In contrast to this, the prediction set in conformal prediction [21] is represented as follows (by using notations equivalent to the original ones for easier comparison):
> > >
> > > $
> > > \Gamma^\epsilon (x) = \\{ y \in \mathcal{Y} \mid p^y(x) \ge \epsilon \\},
> > > $
> > >
> > > where
> > > $
> > > p^y(x) = \frac{\left| \\{ i = 1, \dots, n \mid \alpha_i \le \alpha^y(x) \\} \right| + 1 }{n+1},
> > > $
> > > $\alpha_i = f(x_i, y_i)$, and $\alpha^y(x) = f(x, y)$.
> > >
> > > In this case, given $x$ and an estimated $\epsilon$, we need to compute $p^y(x)$ for each $y$, which requires an iteration over $n$ calibration examples, to check $ \alpha_i \le \alpha^y(x)$ for $i = 1, \dots, n$, thus taking time linear in $n$.
> > > Of course, this algorithm can be simplified, as it is equivalent to the $f(x,y)$ being larger than the appropriate quantile of the $ \alpha_i$-s, which can be precomputed to reduce test-time cost to $O(1)$. In a sense, this is exactly what PS-Binom does; and it appears it was not explicitly stated in the prior work [21].

---

> > > > ### Author Response · Authors · 2022-08-05
> > > > **Additional Information**
> > > >
> > > > To clarify our point, we found two different implementations for inductive conformal prediction (ICP), suggesting $O(n)$ implementation has been used.
> > > >
> > > > The [first one](https://github.com/donlnz/nonconformist/blob/c32760d82c9c4621a9a241b013a6f89e65e06059/nonconformist/icp.py#L255) implemented a prediction set via the p-value approach, as stated in the conformal prediction paper; thus it has $O(n)$ running time. The last code commit date is 2018.
> > > >
> > > > The [second one](https://github.com/aangelopoulos/conformal_classification/blob/db3a42f47d4f3a4cab33bbf577a1257c08366dfd/conformal.py#L35) implemented a prediction set via the quantile approach, thus $O(1)$ running time. The related paper is published in ICLR 2021.
> > > >
> > > >
> > > > Based on this, we observe that the efficient $O(1)$-time implementation can be seen recently (at least 2020 for [3] and 2021 for the second implementation); previously the original $O(n)$-time p-value based prediction set can be used for conformal prediction. Note that they are not implemented for training conditional inductive conformal prediction, but the similar ideas can be applicable.

---

> > > > > ### Comment · Reviewer_e4fH · 2022-08-05
> > > > > **Continuing the clarification**
> > > > >
> > > > > Thanks for the clarifications. I think I understand a bit better now that you are talking about the efficiency of the actual set construction procedure.
> > > > >
> > > > > Another question: the algorithm for constructing ICP prediction sets in $O(1)$ time was known well before [3], right?  For example, Equation 10 of the CQR paper has the $O(1)$ algorithm for constructing the sets.  As far as I know, it was known before that as well.
> > > > >
> > > > > https://arxiv.org/pdf/1905.03222.pdf
> > > > >
> > > > > So [21] has the quantile calculation for how to get training-set-conditional coverage (you take a slightly adjusted quantile from the standard conformal one), and it was already known broadly how to efficiently construct prediction sets given such a quantile. Therefore, I still do not understand the contribution of [3].
> > > > >
> > > > > Can you continue to clarify?

---

> > > > > > ### Author Response · Authors · 2022-08-07
> > > > > > **Clarification**
> > > > > >
> > > > > > Thanks for raising additional concern. To our understanding, ICP prediction set construction in a O(1) time was *not well-known* before. The CQR paper appeared on arXiv in May 2019 and [3] appeared on Openreview in Sept 2019. Based on this, we would say that as conformal prediction got more attention, a simpler representation became widely known only more recently; most likely the authors of [3] did not know this. We will change the wording in the paper to reflect this.
> > > > > >
> > > > > > Of course, we hope as much as the reviewer that the community will come to a common agreement about this, and that every work is properly recognized for its contributions.
> > > > > > In any case, this question about the prior work [3] should have no bearing about our present work at this time. Our contributions should be evaluated in the context of all known work prior to our paper.
> > > > > >
> > > > > > Therefore, please let us know if we have addressed your concerns on clarity and novelty in this paper.

---

> > > > > > > ### Comment · Reviewer_e4fH · 2022-08-07
> > > > > > > **Let's get this right**
> > > > > > >
> > > > > > > "To our understanding, ICP prediction set construction in a $O(1)$ time was not well-known before."
> > > > > > >
> > > > > > > I will be blunt: this could not be farther from the truth. The _original_ ICP paper, which is 20 years old, and cited in the present manuscript, has a whole section on exactly this topic.  It is not buried either, it is in the abstract.  Nearly everyone working in the area has read or at least looked over this manuscript. The authors of [3] may have missed it at the time, but this fact was essentially as _well-known_ as it is possible to be in conformal prediction by 2019. It is totally fundamental.
> > > > > > >
> > > > > > > Here is the paper:  https://citeseerx.ist.psu.edu/viewdoc/download;jsessionid=EBB9348223AD65A77607C337846AF986?doi=10.1.1.4.7849&rep=rep1&type=pdf
> > > > > > >
> > > > > > > (See Section 4, Explicit ICM)
> > > > > > >
> > > > > > > "In any case, this question about the prior work [3] should have no bearing about our present work at this time."
> > > > > > >
> > > > > > > My concern is with the way [3] is attributed in your paper, not with [3] itself. The related work your present paper has to be correct in order for it to be accepted. Peer-review is exactly the mechanism by which "the community", as you say, ensures that proper attribution happens. As the authors stated themselves, every work should be properly recognized for its contributions.
> > > > > > >
> > > > > > > In particular, line 76 says that [3] enables a more efficient algorithm. In light of the above, I hope the authors will see my concern. Furthermore, lines 103-105 are a straw-man, as the $O(1)$ algorithm was well-known prior work.
> > > > > > >
> > > > > > > If these two lines are amended, I will change my score, as I said in the other comment.

---

> > > > > > > > ### Author Response · Authors · 2022-08-07
> > > > > > > > **Thank you for all your feedback!**
> > > > > > > >
> > > > > > > > First, given that it is irrelevant from the perspective of our submission, we have updated our paper, removing the comment about algorithmic efficiency.
> > > > > > > >
> > > > > > > > Next, we would like to clarify our prior comment for our own understanding. As we indicated in a subsequent comment, our perspective was based on the marginal setting vs. the training conditional/PAC setting. When we say we believe the algorithm was not well known before, we were referring to the latter setting. Our understanding is that (Vovk et al, 2013) does not describe how to compute a fixed threshold analogous to the one they compute in the marginal setting. Thus, the efficient algorithm they use in the marginal setting cannot be directly applied in conjunction with their approach for the training conditional/PAC setting. We would be very interested in knowing if we are mistaken!
> > > > > > > >
> > > > > > > > Thank you again for all your feedback!

---

> > > > > > > > > ### Comment · Reviewer_e4fH · 2022-08-08
> > > > > > > > > **Glad I could help**
> > > > > > > > >
> > > > > > > > > Great! I agree that this is irrelevant, and I'll increase my score.
> > > > > > > > >
> > > > > > > > > The threshold in Vovk 2013 is the same as the one in the marginal setting, so I don't think the comment applies.
> > > > > > > > > The fact is that $O(1)$ algorithms for constructing prediction sets were known for 11 years before that paper. The whole point of ICPs is that they are $O(1)$, as opposed to other forms of conformal prediction.
> > > > > > > > >
> > > > > > > > > Vovk et al. sometimes write the full version of conformal down because ICPs are a special case, and they find it notationally convenient to unite the topics. But when in Vovk 2013 he does the experiments, he refers to Vovk 2005 (and previously to Papadopoulos 2002) which contain the computationally efficient version of ICPs, and says it has the advantage of being more computationally efficient than full conformal. In other words, in 2013, it was already obvious how to do this, to the point that those in the community referred to Papadopoulos 2002 for the computationally efficient algorithmic descriptions. Training-conditional ICPs are literally just a special case of ICPs with a slightly adjusted threshold, and it would certainly be unfair to attribute the increase in computational efficiency to [3]. And it is totally unsubstantiated to say that it was "cast[ing] the problem in the language of learning theory" that allowed this increase in computational efficiency. (By the way, the authors may want to look for the full version of the training-conditional ICPs paper, which as I recall also has a section on the connection to PAC theory.)
> > > > > > > > >
> > > > > > > > > I think in the future it would be okay to refer to [3] as the first to do training-conditional conformal prediction in deep learning, and a more convenient/notationally consistent way of aggregating the existing theory. That's definitely a contribution of its own, given how difficult it is to read and interpret works like Vovk 2013, which would be difficult to read and require too much context (since Vovk refers to the existing literature for implementation details of the computationally efficient version of ICPs) for a modern machine learning audience.
> > > > > > > > >
> > > > > > > > > Thanks to the authors for making the requested amendments, and for bearing with this reviewer. I look forward to reading future papers on these topics as the PAC/training-conditional validity is important and under-studied. Glad we could iron out the attribution issues, so they do not appear in the future, and hopefully both of us learned! I did when looking through the references.

---

> > > > > > > > > > ### Author Response · Authors · 2022-08-08
> > > > > > > > > > **Thanks a lot!**
> > > > > > > > > >
> > > > > > > > > > Thanks for the information! We certainly want to get the attribution right as well, and appreciate the detailed response.

---

> > > > > > ### Author Response · Authors · 2022-08-07
> > > > > > **An additional note**
> > > > > >
> > > > > > In addition, we would like to note that CQR is for the marginal setting, where the threshold parameterization is more straightforward (and has been known for some time). Our discussion is regarding the training conditional/PAC setting, where we are not aware of any such efficient approaches -- at least, the efficient approach was not clear to us based on reading (Vovk 2013). We are happy to weaken this discussion, given that it is about prior work rather than our work. In any case, the learning-theoretic viewpoint in [3] was critical for our own understanding of the problem as well as our extension to the meta-learning setting.
> > > > > >
> > > > > > We have updated the paper to say that the efficient approach was clarified by [3] rather than that it was discovered by [3] (highlighted in blue).

---

### Official Review · Reviewer_4tu2 · 2022-07-13

**Rating:** 6
**Confidence:** 4
**Soundness:** 3 good
**Presentation:** 3 good
**Contribution:** 3 good

**Summary:**

This paper considers the notions and algorithms for generating a prediction set under the meta learning setting that satisfies PAC guarantees. Since there are multiple distributions involved in meta learning setting, the considered PAC notion for meta learning is different from existing notion designed for conventional machine learning setting, such as [3]. Furthermore, the meta learning setting is also different from covariate shift case considered in [6]. For example, adaptation data sets are available for a trained model to adapt these associated distributions without estimation of importance weights. Therefore, meta-PAC (mPAC) notion is proposed (Definition 3), which uses an augmented calibration set from N distributions, i.e., $p_{\theta_1}, …, p_{\theta_N}$ to generate $\varepsilon$-correct prediction set for a testing distribution $p_\theta$ whp $1-\delta$. Algorithm 1 is proposed to determine the parameter $\tau$ of prediction set, which uses a two-step strategy to first determine ($\varepsilon, \alpha/2$)-PAC estimator/threshold on each of N distributions, and then use these thresholds (with a label 1 attached to each threshold) to construct a new sample of size N to determine a threshold with ($\alpha/2, \delta$)-PAC. Theorem 1 shows the guarantee of this algorithm for ($\varepsilon, \alpha, \delta$)-mPAC.

**Questions:**

Algorithm 2 involves grid search for $\tau$. Could the authors elaborate a bit more on its computational cost/efficiency in analysis or experiment (e.g., the Granularity for searching)? Would there be any possible way to accelerate such grid search?

**Limitations:**

I think the authors adequately addressed the limitations and potential negative societal impact of their work

**Strengths And Weaknesses:**

Originality:
This paper generalizes PAC notation of conformal prediction from previous studies to mPAC in meta learning case, where multiple distributions are present, augmented calibration set is available. Algorithm is proposed to guarantee mPAC built on the existing algorithm designed for PAC prediction.

Quality:
This paper is technically sound and theoretical analysis is clear.

Clarity:
This paper is easy to follow.

Significance:
This paper provides insights to conformal prediction in meta learning setting.

---

> ### Author Response · Authors · 2022-08-01
> **Response**
>
> We appreciate the reviewer’s interest in our paper! We will share the revised paper soon.
>
> > Question 1. Algorithm 2 involves grid search for τ. Could the authors elaborate a bit more on its computational cost/efficiency in analysis or experiment (e.g., the Granularity for searching)? Would there be any possible way to accelerate such grid search?
>
> Here, we search $\tau$ starting from $0$ and increasing it by the granularity of $10^{-7}$. The computational cost for each iteration is very cheap as long as we first extract scores (i.e., $f(x, y)$) over the calibration set; then, we do not need to execute the neural network again for each candidate $\tau$. The search procedure can be heuristically accelerated via coarse-to-fine search, i.e., initially partition the search space with coarse granularity and search with fine-granularity over one coarse partition. Note that our algorithm can be used with any PAC prediction set algorithm; we simply use the existing Algorithm 2, which conducts grid search.

---

### Official Review · Reviewer_TeH4 · 2022-07-13

**Rating:** 6
**Confidence:** 4
**Soundness:** 3 good
**Presentation:** 3 good
**Contribution:** 3 good

**Summary:**

This paper builds on PAC prediction sets by extending it to few-shot meta-learning settings, where there is assumed access to a i.i.d. sequence of tasks. In few-shot meta-learning, the goal is to adapt to a new task with only a few training examples. In order to hope to do this successfully, the learner is also exposed to a collection of similar tasks (i.e., i.i.d. from some distribution over tasks) before being presented with the test task. From a confidence point of view, providing theoretically valid, non-vacuous confidence sets for few-shot learning is difficult. The work of [1] first proposed a method of also using the task distribution to help derive meaningful conformal prediction sets, that have marginal coverage guarantees over the draw of calibration tasks and test task. This paper adopts this setting, but casts it under the framework of PAC prediction sets, which offer high probability guarantees about the expected mis-coverage over new tasks, conditional on observed calibration tasks. On a high-level, PAC sets is to CP sets as Meta-PAC sets are to Meta-CP sets, and offer a different set of (desireable) performance guarantees. The paper empirically validates the effectiveness (and theoretical properties) of their algorithm on simulated few-shot meta-learning datasets (i.e., K-shot N-way image classification and relation classification tasks), as well as a realistic heart attack prediction task (where the "tasks" are stratified by year and state).

[1] Few-shot conformal prediction with auxiliary tasks. https://arxiv.org/abs/2102.08898.

**Questions:**

- In the experiments it seems that you calibrate using tasks sampled from N choose k base calbration classes, and test on N choose k base (disjoint) test classes. Samples from the second are not i.i.d. w.r.t the second?
 - Lines 171-174 I can't understand. From what I can see, you simply want to estimate a value t s.t. $P(\hat{\tau}_{\alpha/2, \delta}(S) \leq t) \geq 1 - \delta$. I'm not sure what $g$ is, and adding the dummy label also seems confusing.

**Limitations:**

The authors adequately addressed the limitations and potential negative societal impact of their work. It might be worth to expand on the last sentence of the conclusion: namely that people should understand that this algorithm is not a cure-all for few-shot confidence estimation, but one that relies on several strong assumptions, and only holds on average across new tasks. That said, it is better than nothing!

**Strengths And Weaknesses:**

The paper is well-written (for the most part, see below) and the experiments are interesting. As the authors do well in pointing out, their PAC calibration framework offers several desireable guarantees that are not covered by the Meta-CP procedure. For example, Meta-PAC sets will be valid on average across all new tasks, rather than just on the (n + 1)th, marginalized over the draw of tasks {1, ..., n + 1}. Like standard PAC sets, this is nice, as it is often more practical to assume that the same fixed calibration set is reused for predictions across many future tasks. Therefore, even though prediction sets have been considered in a meta-learning setting before, this seems like a meaningful contribution.

Apart from a few specific comments/concerns, my main remark is on the readability of the presentation. Granted, I think the paper does an ok job considering the number of factors to consider simultaneously, but the notation is quite burdensome. It may be much easier to follow if a high-level, step-by-step description of what the authors want to accomplish and how they do it, is provided at the start of section 4---instead of jumping right into notation. For example, at a high-level, Meta-PAC sets are trying to find a single, shared threshold to use that
1. On average over the random draw of calibration tasks that are used to choose this threshold...
2. and on average over the random choice of task that is used to evaluate this threshold...
3. and on average over predictions the model makes using this threshold on the new task...
4. The prediction set contains the correct label.

(Where we can replace "average" with high probability $1-\delta$, $1-\alpha$, $1-\epsilon$.)

I think this message is hinted at in various places in the manuscript, but it would be useful to say it plainly up front (obviously, not exactly in those words, but in simple, intuitive terms).

On to a few high-lighted strengths...

- The theoretical setting is appealing, and a reasonable and useful extension to Meta-CP (as stated above).
- The calibration strategy seems correct and nice. At first I was concerned that computing PAC-thresholds for each of the calibration tasks was missing a factor of 1/N so that they are valid simultaneously, before applying the PAC method again on top of these values---as, with high probability, many $\hat{\tau}_{\epsilon, \alpha/2}(S_i, A_i)$ will _not_ be in $T_\epsilon(\theta_i, A_i)$ for large N. So we'd be calibrating the final threshold based on some "imposter" thresholds from a subset of tasks {1, ..., N}. However, my understanding is that since we only care about the output of a PAC method being i.i.d. (this is the r.v. O = PS-Binom(...) where O is valid w.p. 1 - alpha), not the output being valid itself, we do not actually need the PAC property to hold simulataneously over calibration tasks. Hence, the authors are able to avoid the 1/N factor in favor of two 1/2 factors (which is way more efficient).
- Empirical results seem strong.

And a few weaknesses...

- It's kind of bothersome that a single threshold is used for _all_ downstream tasks; i.e., $\tau$ never gets updated, regardless of the size of the adaptation set A. In contrast, the Meta-CP method attempts to also estimate a data-dependent threshold (i.e., the adjusted quantile prediction). Therefore, when there is enough few-shot data, or significant variability across tasks, this could be better (in terms of being test-task conditionally). Here, the threshold never changes, regardless of whether the new task seems incredibly hard or incredibly easy.
- How tight are Clopper-Pearson based bounds when using small calibration task sizes? Assuming access to several hundred i.i.d. calibration tasks seems rather unrealistic, apart from very specific scenarios. The theory is technically solid, but a relatively straightforward combination of previous techniques (Meta-CP and PAC calibration). Thus I anticipate it to have only a relatively modest impact.

---

> ### Author Response · Authors · 2022-08-01
> **Response**
>
> We appreciate valuable feedback! We will share the revised paper soon.
>
> ---
>
> > It may be much easier to follow if a high-level, step-by-step description of what the authors want to accomplish and how they do it, is provided at the start of section 4---instead of jumping right into notation.
>
> Thanks for the suggestion; we gave the suggested overview of the idea before Section 4.1.
>
> ---
>
> > Weakness 1. It's kind of bothersome that a single threshold is used for all downstream tasks; i.e., τ, never gets updated, regardless of the size of the adaptation set A. In contrast, the Meta-CP method attempts to also estimate a data-dependent threshold (i.e., the adjusted quantile prediction). Therefore, when there is enough few-shot data, or significant variability across tasks, this could be better (in terms of being test-task conditionally). Here, the threshold never changes, regardless of whether the new task seems incredibly hard or incredibly easy.
>
> Intuitively, the threshold is fixed for all future tasks since there is not enough data in the new task to provide much statistical information about the quality of the scoring function in that task, let alone construct separate held-out calibration sets. Instead, our approach relies on meta-learning algorithms to heuristically construct good scoring functions for the new domain, thereby performing well in conjunction with the fixed threshold. While we agree it would be interesting to adapt the threshold, we believe our approach is an effective solution to the problem given the limitations of small datasets from new domains.
>
> ---
>
> > Weakness 2. How tight are Clopper-Pearson based bounds when using small calibration task sizes? Assuming access to several hundred i.i.d. calibration tasks seems rather unrealistic, apart from very specific scenarios. The theory is technically solid, but a relatively straightforward combination of previous techniques (Meta-CP and PAC calibration). Thus I anticipate it to have only a relatively modest impact.
>
> To the best of our knowledge, the Clopper-Pearson bound is the tightest bound for the parameter of a binomial distribution, since it uses the exact tail of the binomial distribution. Of course, the interval size can be large when given a small calibration task size. We believe a few hundred tasks are necessary to satisfy a desired guarantee with reasonable prediction set size. Furthermore, we note that meta-learning often requires more than a hundred tasks (e.g., 200 in [14]), and we believe that most settings where meta-learning is effective satisfy this requirement. The most common task we know of is where new tasks correspond to new labels, and for many domains (e.g., image classification), having hundreds of labels is a reasonable assumption.
>
> ---
>
> > Question 1. In the experiments it seems that you calibrate using tasks sampled from N choose k base calbration classes, and test on N choose k base (disjoint) test classes. Samples from the second are not i.i.d. w.r.t the second?
>
> We are not fully sure whether we understand the question. Here, we assume that tasks are iid and samples from each task are iid. However, a sample from one task and a sample from another task are not drawn from the same distribution (as two tasks are different). In this sense, the two samples from the two different tasks are not iid. Let me know if this addresses your concern.
>
> ---
>
> > Question 2. Lines 171-174 I can't understand. From what I can see, you simply want to estimate a value $t$ s.t. $P(\hat\tau_{\alpha/2, \delta}(S) \le t) \ge 1−\delta$. I'm not sure what g is, and adding the dummy label also seems confusing.
>
> Your point is correct; here, we introduce $g$ and the dummy label to align it with a conventional prediction set construction algorithm. This enabled us to use the same algorithm PS-Binom twice in our method, and thus we thought it makes our presentation concise and elegant. For clarity, we added the equivalent but simpler description that you suggested as well.

---

> > ### Comment · Reviewer_TeH4 · 2022-08-09
> > **Thanks for your response**
> >
> > Thanks for improving the clarity of the presentation.
> >
> > - I agree that there's not too much you can do when the data sample size is small, so fixing t is ok considering the options. That said, you are still assuming that all of the few-shot scores are essentially self-normalizing, right? Suppose in the limit you had an oracle model that had perfect separation between correct and incorrect classes, even given a few data points (e.g., the next large language model). Across tasks/domains, the confidence scores this model gives, even if perfectly separable, would have to be comparable for the same fixed threshold to work. On the other hand, estimating a bias given a few points from the same dataset may not be that difficult. Might be worthwhile to more explicitly discuss this limitation.
> >
> > - My question about i.i.d-ness: the calibration tasks are sampled N-choose-K from a set of shared base classes. I can see that this resulting set of tasks is exchangeable, but are they necessarily i.i.d.? I would think that they are only i.i.d. given the sample of shared base classes? Likewise, since the calibration tasks are sampled from a different subset of N tasks than the test tasks, even if these calibration tasks are i.i.d w.r.t. themselves, are they i.i.d. w.r.t. the test data?
> >
> > Apart from that, I still think the paper has merit, in that it does give a task-conditional meta learning extension to Fisch et. al. I am keeping my original score (as it reflects that). Thanks to reviewer e4fH for straightening out the attribution errors, which I also agree with.

---

> > > ### Author Response · Authors · 2022-08-10
> > > **Thanks for supporting this paper**
> > >
> > > Thanks for supporting this paper!
> > >
> > > **Score normalization**: Note that we do not need to normalize scores to obtain our correctness guarantees. However, as you suggest, when using few-shot learning to adapt the scoring function for each domain, it can be helpful to normalize scores (e.g., by calibrating them using temperature scaling) to improve performance (i.e., reduce the prediction set sizes), though we did not find it necessary in our experiments. We are happy to add a discussion to our paper.
> > >
> > > **IID-ness**: What we have in mind is that there is some (possibly infinite) space $L$ of all possible class labels, and the set of K class labels $\ell_1,...,\ell_K$ are always sampled IID (with replacement) from some distribution over $L$. In this model, there may be overlap between the class labels in the training and test tasks, but it would be tiny as long as $L$ is very large. We believe this is a reasonable model for many real-world applications. Note that in our experiments, we use standard methodology where we use disjoint training/calibration/test label sets, but we argue this is more for experimental best practices rather than realism (also, our experiments show that our approach works well even in this setting).
> > >
> > > We agree that it would be interesting to see whether the i.i.d. assumption can be relaxed to exchangeability while achieving the same conditional guarantee, but believe this question is beyond the scope of our paper. We are happy to add a discussion to our paper.
> > >
> > > We appreciate the valuable comments, which helped a lot to improve our paper.

---

### Author Response · Authors · 2022-08-03
**The first paper revision**

Dear all,

We have uploaded the revised paper based on the initial reviews  (where the revised contents are highlighted in red). We appreciate the constructive comments and will keep revising based on the additional discussion!

---

### Author Response · Authors · 2022-08-05
**The second paper revision**

Dear all,

We have revised our paper for simplifying notations as suggested by Reviewer e4fH during the discussion period; we mainly tried to reduce cognitive cost, e.g., by avoiding the subscript and superscript usages (if possible) and using a different estimator notation for the base PAC prediction set parameter, i.e., $\hat\gamma$ instead of $\hat\tau$.

---

### Meta-Review · Area_Chair_oNiw · 2022-08-29

**Recommendation:** Accept
**Confidence:** Less certain

**Metareview:**

This paper considers the problem of uncertainty quantification in the meta learning setting, where the machine learning model needs to adapt to new classfication tasks. It solves this problem within the framework of conformal prediction to produce PAC prediction sets. This paper generalizes PAC notation of conformal prediction to mPAC in meta learning setting and develops an approach to guarantee mPAC by building on the existing algorithm designed for PAC prediction. Empirical results seem reasonably strong.

There was a lot of discussion for this paper which resulted in a much improved paper in terms of clarity and attribution to prior art. The paper is technically correct and is the second paper in the meta learning setting after Fisch et al. (https://arxiv.org/abs/2102.08898). The difference between Fisch et al. and the proposed approach is that this one applies the training-conditional idea twice in order to condition over tasks and calibration sets. In some sense, it is essentially a more complicated variant of an existing method. As some reviewers mentioned, it is not clear even from the empirical evaluation whether the proposed method performs better than the simple one.

Therefore, I consider this a *borderline paper*. I'm leaning towards accepting (if there is room in the conformal prediction related papers) for the following reason: not many conformal prediction researchers are working on large-scale ML problems and seeing this paper may inspire them to work on such problems.

**Award:**

No

---

### Decision · Program_Chairs · 2022-09-14

Accept